# Optimal Convergence Rate for Exact Policy Mirror Descent in Discounted Markov Decision Processes

**Emmeran Johnson**
Department of Mathematics
Imperial College London
emmeran.johnson17@imperial.ac.uk

**Ciara Pike-Burke**
Department of Mathematics
Imperial College London
c.pike-burke@imperial.ac.uk

**Patrick Rebeschini**
Department of Statistics
University of Oxford
patrick.rebeschini@stats.ox.ac.uk

## Abstract

Policy Mirror Descent (PMD) is a general family of algorithms that covers a wide range of novel and fundamental methods in reinforcement learning. Motivated by the instability of policy iteration (PI) with inexact policy evaluation, PMD algorithmically regularises the policy improvement step of PI. With exact policy evaluation, PI is known to converge linearly with a rate given by the discount factor $\gamma$ of a Markov Decision Process. In this work, we bridge the gap between PI and PMD with exact policy evaluation and show that the dimension-free $\gamma$-rate of PI can be achieved by the general family of unregularised PMD algorithms under an adaptive step-size. We show that both the rate and step-size are unimprovable for PMD: we provide matching lower bounds that demonstrate that the $\gamma$-rate is optimal for PMD methods as well as PI, and that the adaptive step-size is necessary for PMD to achieve it. Our work is the first to relate PMD to rate-optimality and step-size necessity. Our study of the convergence of PMD avoids the use of the performance difference lemma, which leads to a direct analysis of independent interest. We also extend the analysis to the inexact setting and establish the first dimension-optimal sample complexity for unregularised PMD under a generative model, improving upon the best-known result.

## 1 Introduction

The problem of finding an optimal policy in tabular discounted Markov Decision Processes (MDPs) was classically solved using dynamic programming approaches such as policy iteration (PI) and value iteration (VI) [29, 35]. These methods are well understood theoretically and are guaranteed to converge linearly to the optimal policy in the tabular setting with a rate equal to the discount factor $\gamma$ of the MDP [9]. Recently, increased interest has been devoted to the study of policy-gradient (PG) approaches based on optimising a parameterised policy with respect to an objective [36, 23, 20].

Given their popularity, it is of interest to better understand PG methods and determine if their guarantees match those of classical algorithms in tabular MDPs. Among the recent works focused on understanding these methods in the tabular setting, [38] study a general family of algorithms known as Policy Mirror Descent (PMD). PMD algorithmically regularises the policy improvement step of PI and as such can be seen as a version of regularised PI, without actually regularising the objective of interest. It is also viewed as a policy-gradient method through its connection to mirror descent [8]. Linear convergence of PMD was established by [38], though their rate depends on an instance-

dependent factor that can scale with the dimension of the problem, such as the size of the state space. For a specific instance of PMD known as Natural Policy Gradient (NPG), [22] showed that an instance-independent $\gamma$-rate is achievable, although their results do not cover general PMD. In MDPs where the objective is regularised, the $\gamma$-rate has been established for PMD [13, 24, 41]. The classical approaches (PI and VI) achieve the $\gamma$-rate without regularisation, revealing that regularisation is, in general, not necessary for algorithms to reach the $\gamma$-rate. This motivates the following questions:

*Can the classical linear $\gamma$-rate be matched by unregularised policy-gradient algorithms? And what is the best rate that unregularised policy-gradient methods can achieve ?*

For PMD, our work answers the first question positively and answers the second by establishing that the $\gamma$-rate is in fact the best rate achievable for PMD as well as for a more general family of algorithms (see Section 4.1). PMD allows for a choice of a mirror map that specifies different algorithms. Among these, NPG and PI are two ubiquitous instances of PMD each corresponding to their own mirror map. However, PMD is much more general and other mirror maps will lead to alternative algorithms endowed with the guarantees of PMD that we establish in this paper. In particular, the correspondence of mirror maps with exponential families [7] allows us to specify a wealth of valid mirror maps. This illustrates that PMD is a general framework that encompasses a wide range of novel but also fundamental algorithms, and motivates the study of its convergence guarantees. In this work, we make the following contributions and summarise them in Table 1,

- We recover the $\gamma$-rate for the general family of PMD algorithms with exact policy evaluations under an adaptive size (see the third bullet point below). In particular, Theorem 4.1 establishes the following bound in $\ell_\infty$-norm for the value $V^{\pi^k}$ of the policy $\pi^k$ after $k$ iterations of PMD compared to the value $V^{\pi^\star}$ of an optimal policy $\pi^\star$,

$$\|V^{\pi^\star} - V^{\pi^k}\|_\infty \leq \frac{2}{1-\gamma}\gamma^k,$$

  providing guarantees for any starting-state distribution. This matches the rate of VI and PI as well as the best known rates for PMD on regularised MDPs. This is also the first fully dimension-independent linear convergence result for unregularised PMD, by which we mean that there is no dependence on the size of the state space or the action space.

- We provide a matching non-asymptotic lower-bound in Theorem 4.2, establishing the $\gamma$-rate as the optimal rate for PMD methods in early iterations. Our results show that a particular choice of learning rate allows PMD to reach this lower-bound exactly. Note that an asymptotic lower-bound is not possible due to the exact convergence of PI in finite iterations (see Section 4.1 for a discussion of the significance of this non-asymptotic lower-bound).

- The $\gamma$-rate for PMD in Theorem 4.1 relies on an adaptive step-size, where the adaptivity comes from the fact that the step-size depends on the policy at the current iteration (see Section 4). In Theorem 4.3 we show that this adaptivity is necessary for PMD to achieve the $\gamma$-rate, establishing our step-size as both sufficient and necessary.

- We establish a novel theoretical analysis that avoids the use of the performance difference lemma [19]. This leads to a simple analysis and avoids needing to deal with visitation distribution mismatches that lead to dimension dependence in prior work.

- By extending our analysis to the inexact setting, with an approach similar to that of in [38], we establish an instance-independent sample complexity of $\tilde{O}(|\mathcal{S}||\mathcal{A}|(1-\gamma)^{-8}\varepsilon^{-2})$ under a generative model, where the notation $\tilde{O}()$ hides poly-logarithmic factors, $\mathcal{S}$ is the state space of the MDP, $\mathcal{A}$ is the action space and $\varepsilon$ is the required accuracy. This improves on the previous best known sample complexity for PMD by removing the dependence on a distribution mismatch coefficient that can scale with problem-dependent quantities such as the size of the state space. More generally, we highlight that the analysis we establish in the exact setting can easily be combined with any other scheme for estimating the Q functions (see Section 5), paving the way for further improvements in instance-independent sample complexity results should more efficient estimation procedures be developed.

Our contributions are primarily on establishing the optimal rate for general (not just NPG) *exact* PMD where we assume access to the true action-values of policies, and the simplicity of the analysis. The sample complexity result in the inexact setting illustrates how our analysis can be easily extended to

Table 1: Comparison of contributions with prior work that study PMD. [22] focus on NPG, an instance of PMD for a specific mirror map (see Section 3). Their analysis is fundamentally different to ours as it exploits the closed-form update of NPG. Their step-size is similar to ours, although it has a dependence on a sub-optimality gap (see Section 4). Linear $\gamma$-rate holds if the linear convergence is with the $\gamma$-rate, not only linear convergence. The $\ell_\infty$-bound is satisfied if it holds for $\|V^{\pi^\star} - V^{\pi^k}\|_\infty$. Dimension independence is satisfied when there is no instance for which the bound can scale with the size of the state or action space. We compare these works in more detail in Section 4.

| | Linear $\gamma$-Rate | General Mirror Map | $\ell_\infty$ Bound | Dimension Independent | Matching Lower-Bound | Step-Size Necessity |
|---|---|---|---|---|---|---|
| [22] | ✓ | × | ✓ | ✓ | × | × |
| [38] | × | ✓ | × | × | × | × |
| This work | ✓ | ✓ | ✓ | ✓ | ✓ | ✓ |

obtain improved results for inexact PMD. These contributions advance the theoretical understanding of PMD and rule out searching for instances of PMD that may improve on PI or NPG in the exact setting. The algorithmic differences between instances of PMD do not affect the performance of the algorithm beyond the step-size condition.

## 2 Related work

### 2.1 Convergence rates for exact policy mirror descent

We first consider the setting where exact policy evaluation is assumed. In this setting, several earlier works have sub-linear convergence results for PMD [17, 33] and NPG specifically [3, 1], though these have since been improved to linear convergence results as discussed below. Note that the work of [1] refers to their algorithm as "Politex", which shares the same update as NPG.

A line of work has considered PG methods applied to regularised MDPs. In this setting, linear convergence has been established for NPG with entropy regularisation [13], PMD with strongly-convex regularisers [24] and PMD with convex non-smooth regularisers [41]. The rates of convergence are either exactly $\gamma$ or can be made arbitrarily close to $\gamma$ by letting the step-size go to infinity.

In the setting of unregularised MDPs, which is the focus of this paper, linear convergence of the special case of NPG was established [11, 22] under an adaptive step-size similar to ours that depends on the current policy at each step. The bound of [11] has an additive asymptotic error-floor that can be made arbitrarily small by making the step-size larger, while for a similar step-size [22] does not have this term so we focus on this work. Their analysis relies on a link between NPG and PI and consists of bounding the difference in value between iterates of both methods. [11] also establish linear convergence for a number of algorithms including PMD, although it is in the idealised setting of choosing the step size at each iteration that leads to the largest increase in value. This step-size choice will make PMD at least as good as PI since arbitrarily large step-sizes can be chosen and PMD with an infinite step-size converges to a PI update. Since PI converges linearly, so will PMD. This does not establish linear convergence of PMD for step-sizes with a closed-form expression. However, linear convergence for unregularised general PMD was recently established by [38] under a geometrically increasing step-size. In general, their rate is instance-dependent and may scale with problem dependent quantities such as the size of the state space. For general starting-state distributions, this same instance-dependent rate was established by [27] for a variant of PMD which augments the update with an added regularisation term. We focus our comparison on the work of [38] rather than this work as the guarantees are equivalent in both but [27] do not directly study PMD and have a more complicated algorithm. A summary of our results compared to those of [22] and [38] is presented in Table 1 and discussed in more detail in Section 4. In terms of optimality, [22] provide a lower-bound for constant step-size NPG, though it only applies to MDPs with a single-state, which can be solved in a single iteration with exact policy evaluation as the step-size goes to infinity (for which the lower-bound goes to 0). We provide a lower-bound in Theorem 4.2 that applies to PMD with arbitrary step-size on an MDP with any finite state space. To the best of our knowledge, prior to this work no lower-bound has been established in this general setting.

## 2.2 Sample complexity of inexact policy mirror descent

Sample complexity in the inexact policy evaluation setting refers to the number of samples needed to guarantee an $\varepsilon$-optimal policy is returned when we no longer have access to the Q-values exactly. We here give an outline of results, typically established in high-probability, under a generative model that we formally present in Section 5. The lower bound on the sample complexity in this setting was shown to be of $\tilde{\Omega}\left(\frac{|\mathcal{S}||\mathcal{A}|}{(1-\gamma)^3\varepsilon^2}\right)$ by [6]. This lower-bound can be reached by model-based approches [2, 26] and model-free approaches [34, 37].

The sample-complexity for PG methods has been recently studied in [39]. Under a generative model, some works have considered PMD or NPG under various types of regularisation [13, 24, 41]. We focus on unregularised methods, for which results for PMD or its instances on tabular MDPs under a generative model are limited. There are works that obtain sample complexity results for NPG [3, 28] and for PMD [33] though they do not attain the optimal $\varepsilon$-dependence of $\mathcal{O}(\varepsilon^{-2})$. [25] show that a variant of PI, a special case of PMD, achieves the optimal $\varepsilon$-dependence of $\mathcal{O}(\varepsilon^{-2})$. More recently, [38] show that the general family of PMD methods match the $\mathcal{O}(\varepsilon^{-2})$ sample complexity with a factor of $(1-\gamma)^{-8}$. Our result for the inexact setting shares the same dependence on $\varepsilon$ and $1-\gamma$ as [38] but removes an instance-dependent quantity which can depend on the size of the state space. Further comparison to the result in [38] is given in Section 5. Beyond tabular MDPs and generative models, [15] study NPG under linear function approximation and off-policy sampling, though their results imply worse sample complexities when restricted to tabular MDPs under a generative model. PMD under linear function approximation [40, 4] and general function approximation [5] have also been studied and results similar to [38] were obtained in those settings.

## 3 Preliminaries

A Markov Decision Process (MDP) is a discrete-time stochastic process, comprised of a set of states $\mathcal{S}$, a set of actions $\mathcal{A}$, a discount factor $\gamma \in [0,1)$ and for each state-action pair $(s,a)$ a next-state transition function $p(\cdot|s,a) \in \Delta(\mathcal{S})$ and a (assumed here deterministic) reward function $r(s,a) \in [0,1]$. $\Delta(\mathcal{X})$ denotes the probability simplex over a set $\mathcal{X}$. We consider both $\mathcal{S}$ and $\mathcal{A}$ to be finite, which is known as the tabular setting. In a state $s$, an agent chooses an action $a$, which gives them a reward $r(s,a)$ and transitions them to a new state according to the transition function $p(\cdot|s,a)$. Once they are in a new state, the process continues. The actions chosen by an agent are formalised through policies. A policy $\pi : \mathcal{S} \to \Delta(\mathcal{A})$ is a mapping from a state to a distribution over actions. We will often write it as an element in $\Pi = \Delta(\mathcal{A})^{|\mathcal{S}|}$. In each state $s \in \mathcal{S}$, an agent following policy $\pi$ chooses an action $a \in \mathcal{A}$ according to $\pi_s = \pi(\cdot|s) \in \Delta(\mathcal{A})$.

In this work, the goal is to learn how to behave in a $\gamma$-discounted infinite-horizon MDP. We measure the performance of a policy with respect to the value function $V^\pi : \mathcal{S} \to \mathbb{R}$,

$$V^\pi(s) = \mathbb{E}\Big[\sum_{t=0}^\infty \gamma^t r(s_t, a_t)|\pi, s_0 = s\Big],$$

where $s_t, a_t$ are the state and action in time-step $t$ and the expectation is with respect to both the randomness in the transitions and the choice of actions under policy $\pi$. This is a notion of long-term reward that describes the discounted rewards accumulated over future time-steps when following policy $\pi$ and starting in state $s$. For a distribution over starting states $\rho \in \Delta(\mathcal{S})$, we write $V^\pi(\rho) = \sum_{s\in\mathcal{S}} \rho(s)V^\pi(s)$ for the expected value when starting in a state distributed according to $\rho$. It is also useful to work with the state-action value $Q^\pi : \mathcal{S} \times \mathcal{A} \to \mathbb{R}$:

$$Q^\pi(s,a) = \mathbb{E}\Big[\sum_{t=0}^\infty \gamma^t r(s_t, a_t)|\pi, s_0 = s, a_0 = a\Big],$$

which is similar to $V^\pi$, with the additional constraint of taking action $a$ in the first time-step. We will often write $V^\pi \in \mathbb{R}^{|\mathcal{S}|}$ (resp. $Q^\pi \in \mathbb{R}^{|\mathcal{S}|\times|\mathcal{A}|}$) to refer to the vector form, where each entry represents the value (resp. action-value) in that state (resp. state-action pair). Similarly, we write $Q_s^\pi \in \mathbb{R}^{|\mathcal{A}|}$ for the vector of action-values in state $s$. The following expressions, which relate $Q^\pi$ and $V^\pi$ in terms of each other and when combined give the Bellman equations [9], follow from their definitions above,

$$V^\pi(s) = \langle Q_s^\pi, \pi_s\rangle, \quad Q^\pi(s,a) = r(s,a) + \gamma\sum_{s'\in\mathcal{S}} p(s'|s,a)V^\pi(s').$$

We now define the discounted visitation-distribution for starting state $s'$ and policy $\pi$,

$$d_{s'}^\pi(s) = (1 - \gamma) \sum_{t=0}^\infty \gamma^t \mathbb{P}^\pi(s_t = s | s_0 = s'), \tag{1}$$

which plays an important part in the study of PG methods. Note that $\mathbb{P}^\pi(s_t = s | s_0 = s')$ is the probability of being in state $s$ at time $t$ when starting in state $s'$ and following policy $\pi$. We also write $d_\rho^\pi(s) = \sum_{s' \in \mathcal{S}} \rho(s') d_{s'}^\pi(s)$.

One of the main aims of reinforcement learning is to find a policy $\pi$ that maximises $V^\pi$. It is known that there exists a deterministic policy that simultaneously maximises $V^\pi$ and $Q^\pi$ for all states and actions [9]. We call such a policy an optimal policy and denote it by $\pi^\star$. We are interested in finding an $\varepsilon$-optimal policy, i.e a policy $\pi$ such that $\|V^{\pi^\star} - V^\pi\|_\infty < \varepsilon$.

## 3.1 Exact policy mirror descent

We are interested in PG methods that are based on optimising a parameterised policy $\pi_\theta$ with respect to $V^{\pi_\theta}(\rho)$ for some $\rho \in \Delta(\mathcal{S})$. In the tabular setting, we can use the direct parameterisation of a policy $\pi_\theta$, which associates a parameter to each state-action pair, i.e. we have $\pi_\theta(a|s) = \theta_{s,a}$. We will drop the subscript $\theta$ for notational convenience. The gradient of the value function with respect to this parameterisation [36] is given for each state-action pair (s,a) by

$$\frac{\partial}{\partial \pi(a|s)} V^\pi(\rho) = \frac{1}{1 - \gamma} d_\rho^\pi(s) Q^\pi(s, a). \tag{2}$$

Mirror Descent (MD, [8]) carries out gradient descent in a geometry that is non-Euclidean. Using $-V^\pi(\rho)$ as the minimising objective, the proximal perspective of MD gives an update of the form

$$\pi^{k+1} = \text{argmin}_{p \in \Pi} \left\{ -\eta_k \langle \nabla V^{\pi^k}(\rho), p \rangle + D_h(p, \pi^k) \right\} \tag{3}$$

where $h : \text{dom } h \to \mathbb{R}$ is the mirror map (with $\Pi \subset \text{dom } h$) and $D_h$ is the Bregman divergence generated by $h$. We require $h$ to be of Legendre type [30], i.e strictly convex and essentially smooth (differentiable and $\|\nabla h(x_k)\| \to \infty$ for any sequence $x_k$ converging to a point on the boundary of dom $h$) on the relative interior (rint) of dom $h$. The Bregman Divergence is defined as

$$D_h(\pi, \pi') = h(\pi) - h(\pi') - \langle \nabla h(\pi'), \pi - \pi' \rangle \qquad \text{for } \pi, \pi' \in \text{dom } h.$$

As the objective $V^\pi(\rho)$ is non-convex in general [3], usual techniques from convex theory [12] are not applicable.

The presence of the visitation-distribution term $d_\rho^\pi(s)$ in the gradient of the objective in (2) can slow down learning because it can lead to vanishingly small gradients when states are infrequently visited under the current policy $\pi$ [3]. To circumvent this issue, Policy Mirror Descent (PMD) [24, 33, 38] applies a variant of update (3) with a weighted Bregman divergence $D_h^{\text{PMD}}$ that matches the visitation distribution factors of the gradient $D_h^{\text{PMD}}(p, \pi^k) = \sum_s d_\rho^{\pi^k}(s) D_h(p_s, \pi_s^k)$ where the mirror map $h$ is now defined on a subset of $\mathbb{R}^{|\mathcal{A}|}$. The resulting update has for all states a factor of $d_\rho^{\pi^k}(s)$ in both terms. The minimisation can then be applied for each state individually to get the PMD update

$$\pi_s^{k+1} = \text{argmin}_{p \in \Delta(\mathcal{A})} \left\{ -\eta_k \langle Q_s^{\pi^k}, p \rangle + D_h(p, \pi_s^k) \right\} \tag{4}$$

for all states $s$. We will often add a superscript $k$ to any quantity that is associated to $\pi^k$. For example, $V^k(s) = V^{\pi^k}(s)$. Similarly for $\pi^\star$ and the superscript $\star$. Exact PMD iteratively applies update (4) for some sequence of step-sizes $\eta_k > 0$ and initial policy $\pi^0 \in \text{rint } \Pi$. We call this algorithm exact because we assume access to the true state-action values $Q^k$.

The update (4) of PMD considered in this work uses the true action-value $Q^\pi$. In prior work, PMD has sometimes been applied to regularised MDPs [24] where the action-value is augmented with regularisation and is no longer the true action-value. This is a different algorithm that converges to a policy that is optimal in the regularised MDP but not in the original unregularised MDP.

PMD is a general family that covers many algorithms, specified by the choice of mirror map $h$. These will inherit the guarantees of PMD, which motivates the study of the convergence guarantees of

PMD beyond specific instances. Taking $h$ to be the negative entropy yields NPG, whose theoretical properties have attracted a lot of interest [3, 13, 22]. With a null Bregman Divergence, PMD recovers PI. PI is generated by a constant mirror map, which is not of Legendre type but the analysis still applies so all results on PMD remain valid for PI. In fact, PMD can be viewed as a form of regularised PI since the update (4) converges to a PI update as $\eta_k \to \infty$, regardless of the mirror map. Beyond these, providing mirror maps that generate other Bregman Divergences will lead to different algorithms. In particular, every exponential family has a corresponding mirror map generating a unique Bregman Divergence [7], highlighting the generality of PMD.

## 4 Main results for exact policy mirror descent

In this section, we present our main results on the convergence of exact PMD. We first introduce some relevant notation. Fix a state $s \in \mathcal{S}$ and an integer $k \geq 0$. Let $\mathcal{A}_s^k = \{a \in \mathcal{A} : Q^k(s, a) = \max_{a' \in \mathcal{A}} Q^k(s, a')\}$ denote the set of optimal actions in state $s$ under policy $\pi^k$. Denote by $\widetilde{\Pi}_s^{k+1}$ the set of greedy policies w.r.t $Q_s^k$ in state s, i.e $\widetilde{\Pi}_s^{k+1} = \left\{ p \in \Delta(\mathcal{A}) : \sum_{a \in \mathcal{A}_s^k} p(a) = 1 \right\}$. We are now ready to state our main result in the setting of exact PMD, which is proved in Section 6.

**Theorem 4.1.** *Let $\{c_k\}_{k \in \mathbb{Z}_{\geq 0}}$ be a sequence of positive reals. Consider applying iterative updates of (4) with $\pi^0 \in rint \, \Pi$ and step-sizes satisfying for all $k \geq 0$,*

$$\eta_k \geq \frac{1}{c_k} \max_{s \in \mathcal{S}} \left\{ \min_{\widetilde{\pi}_s^{k+1} \in \widetilde{\Pi}_s^{k+1}} D_h(\widetilde{\pi}_s^{k+1}, \pi_s^k) \right\}. \tag{5}$$

*Then we have for all $k \geq 0$,*

$$\|V^\star - V^k\|_\infty \leq \gamma^k \Big( \|V^\star - V^0\|_\infty + \sum_{i=1}^k \gamma^{-i} c_{i-1} \Big). \tag{6}$$

The sequence $\{c_k\}_{k \in \mathbb{Z}_{\geq 0}}$ plays an important role in both the step-size constraint (5) and the bound (6). In particular, different choices will lead to different guarantees. We focus on $c_i = \gamma^{2(i+1)} c_0$ for some $c_0 > 0$, giving a step-size with a geometrically increasing component. The resulting bound is

$$\|V^\star - V^k\|_\infty \leq \gamma^k \Big( \|V^\star - V^0\|_\infty + \frac{c_0}{1 - \gamma} \Big),$$

which converges linearly with the $\gamma$-rate, and matches the bounds of PI and VI as $c_0$ goes to 0. PMD cannot do better as we will show in Theorem 4.2. We discuss other choices of $\{c_k\}$ in Appendix C. We provide some simulations that demonstrate the behaviour of our step-size (5) and validate its benefits compared to the step-size of [38] in Appendix D.

**Comparison to [38]:** Linear convergence of unregularised PMD was first established by [38] under a geometrically increasing step-size $\eta_k = \eta_0/\gamma^k$. We discuss this step-size further and show the necessity of adaptivity to achieve the $\gamma$-rate in Section 4.2. Their rate of convergence is $1 - \frac{1}{\theta_\rho}$ where $\theta_\rho$ is an instance-dependent term defined as follows

$$\theta_\rho = \frac{1}{1 - \gamma} \Big\| \frac{d_\rho^\star}{\rho} \Big\|_\infty,$$

where $d_\rho^\star$ is the visitation distribution defined in (1) under an optimal policy and $\rho$ is the starting-state distribution to which the bound applies, i.e the bound is on $V^\star(\rho) - V^k(\rho)$. This $\theta_\rho$ is at best $\gamma$ when we use $\rho$ to be the stationary distribution of the optimal policy. However, in this case, the guarantee only applies to states on the support of this stationary distribution and provides no guarantees for other states. In general, it is unclear how $\theta_\rho$ may scale in a specific MDP. In particular, it is possible to construct an MDP where $\theta_\rho$ scales linearly with the size of the state space $|\mathcal{S}|$ (Appendix H.1). Though this MDP is somewhat trivial, it nonetheless illustrates how $\theta_\rho$ can easily be large leading to slow rates of convergence. It is also not straightforward to obtain convergence in individual states from the bound in [38] due to the presence of $\rho$ in the denominator of the mismatch coefficient in $\theta_\rho$. In contrast, we obtain the optimal $\gamma$-rate of convergence and our result holds in $\ell_\infty$-norm over all states so avoids having to deal with a starting-state distribution $\rho$ altogether.

This distribution mismatch coefficient commonly appears in convergence bounds in the literature [19, 32, 10, 33, 3], both under exact and inexact policy evaluation. For many of these papers mentioned, removing it would be of great interest though often does not appear possible. Our results show that it is removable for the general family of PMD algorithms to obtain dimension-free linear convergence. The techniques we use may be of interest for removing this coefficient in other settings.

**Comparison to [22]:** The $\gamma$-rate was established by [22] for NPG, a specific instance of PMD for which the Bregman Divergence is the KL-divergence. The bound shown in their work is similar to the one implied by our result with $c_i = \gamma^{2(i+1)}c_0$. Defining $\Delta^k(s) = \max_{a \in \mathcal{A}} Q^k(s,a) - \max_{a \notin \mathcal{A}_s^k} Q^k(s,a)$, the minimal sub-optimality gap in state $s$ under $\pi^k$, then the step-size corresponding to their bound with the KL as Bregman Divergence is

$$\eta_k \geq \max_{s, \widetilde{\pi}_s^{k+1} \in \widetilde{\Pi}_s^{k+1}} \left\{ \left( L_k + \log|\mathcal{A}| + D(\widetilde{\pi}_s^{k+1}, \pi_s^k) \right) \frac{1}{\Delta_k(s)} \right) \right\},$$

where $L_k = Lk$ for some constant $L > 0$. This highlights the connection with our step-size condition (5). In particular, they both have an adaptive component that depends linearly on the Bregman divergence between the current policy and the greedy policy and a non-adaptive component on which the bound depends. An important difference is that our step-size is independent of the sub-optimality gap $\Delta_k(s)$, and will be robust to situations where this gap is small. We can construct a general family of MDPs for which we can make $\Delta_k(s)$ arbitrarily small and the step-size of [22] will correspondingly become arbitrarily large (Appendix H.2). Despite the apparent similarities with our results, their analysis is significantly different to ours as it exploits the specific closed-form update of NPG to bound the difference in value with an update of PI. Our analysis applies to PMD for a general mirror map (not just NPG) and as such does not utilize specific properties of the mirror map and does not require the analytic solution of the update to be known. Our analysis also easily extends to inexact PMD (see Section 5), which theirs does not.

**Computing the step-size:** The expression (5) for the step-size can be simplified. The minimum over $\widetilde{\pi}_s^{k+1} \in \widetilde{\Pi}_s^{k+1}$ can be removed by taking any $\widetilde{\pi}_s^{k+1} \in \widetilde{\Pi}_s^{k+1}$. It was stated with the minimum to have the smallest condition but that is not necessary. In fact, there will often only be one greedy action, i.e. $\mathcal{A}_s^k$ will contain a single action and there will just be one policy in $\widetilde{\Pi}_s^{k+1}$. As for the maximum over $s \in \mathcal{S}$, this condition is imposed because we are using the same step-size in all states for simplicity. If we allow different step-sizes in each state, then the step-size in state $s$, denoted $\eta_k(s)$ would just have to satisfy (choosing any $\widetilde{\pi}_s^{k+1} \in \widetilde{\Pi}_s^{k+1}$)

$$\eta_k(s) \geq \frac{1}{c_k} D_h(\widetilde{\pi}_s^{k+1}, \pi_s^k).$$

The computational complexity of the step-size is then that of the Bregman divergence between two policies. Note that the step-size (5) is always finite. It can be unbounded when $h$ is of Legendre type on the relative interior of $\Delta(\mathcal{A})$ (as for the negative entropy) but then all iterates will belong to the relative interior of $\Delta(\mathcal{A})$ and the Bregman divergences are well defined so the step-size is finite.

## 4.1 Optimality of PMD

We have established in Theorem 4.1 that PMD achieves a linear $\gamma$-rate. The following result shows that this rate is in fact optimal in a worst-case sense. The proof can be found in Appendix E.

**Theorem 4.2.** *Fix $n > 0$. There exists a class of MDPs with $|\mathcal{S}| = 2n + 1$ and a policy $\pi^0 \in \text{rint} \, \Pi$ such that running iterative updates of (4) for any positive step-size regime, we have for $k < n$:*

$$\|V^\star - V^k\|_\infty \geq \frac{1}{2}\gamma^k \|V^\star - V^0\|_\infty. \tag{7}$$

A key feature of this result is that the bound holds for $k < n$. For a fixed iteration budget, Theorem 4.2 implies that there exists an MDP on which PMD will not do better than the linear $\gamma$-rate for any step-size. The $\gamma$-rate for PMD that we prove in Theorem 4.1 is optimal in this sense.

**Lower-bound beyond k < n?** For a fixed state-space size $|\mathcal{S}| = 2n + 1$, it is not known what lower-bound holds when $k \geq n$. However, PI is an instance of PMD (see Section 3.1) and thus a lower bound that scales with $\gamma^k$ for all $k > 0$ cannot hold since PI converges exactly in finite-iterations

(in fact with a number of iterations that scales linearly with $|\mathcal{S}|$ [31, Theorem 3]). However, an asymptotic lower-bound for PMD under arbitrary step-size will not hold even if we exclude PI from the class of considered algorithms. As the step-size tends to infinity, any PMD update recovers a PI update. This implies that general PMD can be arbitrarily close to PI's exact convergence for the same finite number of iterations. Thus, any lower-bound on the convergence of PMD must be limited to finite iterations. Since the finite iteration convergence of PI only scales linearly with $|\mathcal{S}|$ [31, Theorem 3], the number of iterations guaranteed by Theorem 4.2 has the same dependence on $|\mathcal{S}|$ as the number of iterations needed for exact convergence of PI. To the best of our knowledge, this lower bound on the value convergence of PMD scaling with $\gamma^k$ is new. We expect this result may have been known for the special case of PI, though we could not find a proof of it in the literature. The works that establish a lower bound for PI do so in the setting of exact convergence to the optimal policy [18], not $\varepsilon$-accurate convergence, and for undiscounted MDPs [16].

**Rate-Optimality**   Theorem 4.2 establishes that the $\gamma$-rate is optimal in the first $|\mathcal{S}|/2$ iterations. This non-asymptotic optimality is justified by the discussion above, which highlights that a rate for PMD under arbitrary step-size can only be optimal for a finite number of iterations, and specifically up until the exact convergence of PI.

There are some results on the super-linear convergence of NPG in the literature, though these apply once you have a policy within some neighbourhood of the optimal policy or value. [13] establish such a result for NPG in the regularised case, and [22] in the unregularised case under additional conditions. Theorem 4.2 does not contradict this latter result as for the MDP considered in the proof, the super-linear convergence would kick-in for iterations beyond the $k < n$ considered here.

**Lower-bound for general PG methods:**   The lower bound of Theorem 4.2 in fact applies to any algorithm that at each iteration increases the probability of the current greedy action. The greedy action is the action with highest action-value for the current policy. This covers algorithms more general than PMD and in particular, includes the vanilla PG algorithm.

## 4.2   Adaptive step-size necessity

We have established in Theorem 4.1 that PMD under an adaptive step-size achieves a linear $\gamma$-rate and in Theorem 4.2 that this rate is optimal for PMD. We now show the adaptivity is in fact necessary to achieve the $\gamma$-rate. This strengthens the notion of optimality from the previous section - both the rate and step-size are unimprovable. The proof can be found in Appendix F.

**Theorem 4.3.** *Fix $n > 0$ and $\gamma > 0.2$. There exists an MDP with state-space of size $|\mathcal{S}| = 2n + 1$ and a policy $\pi^0 \in \text{rint } \Pi$ such that running iterative updates of NPG (PMD with h as the negative entropy) that satisfy $\|V^\star - V^k\|_\infty \leq \gamma^k(\|V^\star - V^0\|_\infty + \frac{1-\gamma}{8})$ requires*

$$\eta_{k_i} \geq KL(\tilde{\pi}_{s_i}^{k_i+1}, \pi_{s_i}^{k_i})/2\gamma^{k_i}, \qquad \tilde{\pi}_{s_i}^{k_i+1} \in \widetilde{\Pi}_s^{k+1} \tag{8}$$

*for $i = 1, ..., n$ s.t $k_1 < k_2 < ... < k_n$ where $\{s_1, ..., s_n\}$ are distinct states of the considered MDP.*

This theorem states that there are at least $n$ distinct iterations with $n$ distinct states where the step-size has to be bigger than a quantity depending on the Bregman divergence between the current policy and its greedy policy in the considered state in order to achieve a linear $\gamma$-rate. This is precisely the notion of adaptivity that appears in the step-size condition of Theorem 4.1 and [22]. Theorem 4.3 shows we cannot improve on this in general and provides justification for using an adaptive step-size instead of the one from [38]. Beyond its necessity, the adaptivity of our step-size can be a strength: it is large when needed, small when not. This is illustrated by the simulations we provide in Appendix D.

Theorem 4.3 only holds for the particular case of the negative entropy (NPG). This limitation is due to technical reasons rather than fundamental ones. The proof relies on the closed form update of NPG and we don't have this for general mirror map (see Appendix F). However, this result for NPG is enough to establish the necessity of the step-size for PMD in general.

# 5 Sample complexity of inexact policy mirror descent under generative model

In the previous sections, we have assumed access to the state-action values $Q_s^k$ to carry out the PMD update. In Inexact PMD (IPMD), we replace $Q_s^k$ with an estimate $\widehat{Q}_s^k$ giving the update

$$\pi_s^{k+1} = \text{argmin}_{p \in \Delta(\mathcal{A})} \Big\{ - \eta_k \langle \widehat{Q}_s^k, p \rangle + D_h(p, \pi_s^k) \Big\}. \tag{9}$$

Similarly to the exact case, IPMD iteratively applies update (9) for some sequence of $\eta_k > 0$ and initial policy $\pi^0 \in \text{rint } \Pi$, this time only assuming access to an inexact estimator of $Q^k$.

We consider the setting of a generative model [21], which is a sampling model where we can draw samples from the transition probabilities $p(\cdot|s,a)$ for any pair $(s,a)$. We borrow an estimator common in the literature (see e.g. [38], [25]): for all state-actions pairs $(s,a)$, draw $M_k$ trajectories of length or horizon $H$, i.e samples of the form $\big((s_0^{(i)}, a_0^{(i)}), (s_1^{(i)}, a_1^{(i)}), ..., (s_{H-1}^{(i)}, a_{H-1}^{(i)})\big)_{i=1,...,M_k}$, where $a_t^{(i)}$ is drawn from $\pi^k(\cdot|s_t^{(i)})$, $s_{t+1}^{(i)}$ is drawn from $p(\cdot|s_t^{(i)}, a_t^{(i)})$ and $(s_0^{(i)}, a_0^{(i)}) = (s,a)$. Using these samples, we can define a truncated Monte-Carlo estimate of the values as follows,

$$\widehat{Q}^k(s,a) = \frac{1}{M_k} \sum_{i=1}^{M_k} \widehat{Q}_{(i)}^k(s,a), \quad \text{where} \quad \widehat{Q}_{(i)}^k(s,a) = \sum_{t=0}^{H-1} \gamma^t r(s_t^{(i)}, a_t^{(i)}). \tag{10}$$

We use these $\widehat{Q}^k(s,a)$ to replace $Q^k(s,a)$ in the PMD update step. [38] present a bound on the accuracy of this estimator which is restated in Appendix G. Following the same ideas as [38], we can extend Theorem 4.1 to the inexact setting. The following theorem establishes a sample complexity result, which is the sufficient number of calls to the generative model to obtain an $\varepsilon$-optimal policy. For simplicity, we focus on the step-size following from the choice $c_i = \gamma^{2(i+1)}$.

**Theorem 5.1.** *Consider applying iterative updates of (9) using the Q-estimator in (10) given access to a generative model with $\pi^0 \in \text{rint } \Pi$ and step-sizes satisfying for all $k \geq 0$ (with the definitions of $\mathcal{A}_s^k$ and $\widetilde{\Pi}_s^{k+1}$ suitably adjusted with $\widehat{Q}_s^k$ instead of $Q_s^k$),*

$$\eta_k \geq \max_{s \in \mathcal{S}} \Big\{ \min_{\widetilde{\pi}_s^{k+1} \in \widetilde{\Pi}_s^{k+1}} \frac{D(\widetilde{\pi}_s^{k+1}, \pi_s^k)}{\gamma^{2k+1}} \Big\}.$$

*Fix $\varepsilon > 0$. For any $\delta \in (0,1)$, suppose the following are satisfied for all $k \geq 0$,*

$$K > \frac{1}{1-\gamma} \log \frac{4}{(1-\gamma)\varepsilon}, \quad H \geq \frac{1}{1-\gamma} \log \frac{16}{(1-\gamma)^3\varepsilon} \quad \text{and} \quad M_k = M \geq \frac{\gamma^{-2H}}{2} \log \frac{2K|\mathcal{S}||\mathcal{A}|}{\delta}.$$

*Then with probability at least $1 - \delta$, $\|V^\star - V^k\|_\infty \leq \gamma^k \Big( \|V^\star - V^0\|_\infty + \frac{1}{1-\gamma} \Big) + \frac{8}{(1-\gamma)^3} \gamma^H < \varepsilon$. Choosing $K$, $H$ and $M$ to be tight to their lower-bounds, the corresponding sample complexity is $\tilde{O}\Big( \frac{|\mathcal{S}||\mathcal{A}|}{(1-\gamma)^8\varepsilon^2} \Big)$, where the notation $\tilde{O}()$ hides poly-logarithmic factors.*

The proof can be found in Appendix G.1. The sample complexity established by [38] (Theorem 16) under a generative model and the same Q-estimator is $\tilde{O}\Big( \frac{|\mathcal{S}||\mathcal{A}|}{(1-\gamma)^8\varepsilon^2} \big\| \frac{d_\rho^\star}{\rho} \big\|_\infty^3 \Big)$.

In their work, [38] stresses the interest in reducing the dependence on both $1/(1-\gamma)$ and the distribution mismatch coefficient in order to scale the PMD guarantees to more relevant settings such as function approximation. Theorem 5.1 partially resolves this matter by removing the dependence on the distribution mismatch coefficient, which may scale with the size of the state space (Appendix H.1). This makes the result dimension-optimal, which is crucial when scaling the results to large or infinite state or action spaces. The dependence on $1/(1-\gamma)$ remains distant from the $1/(1-\gamma)^3$ lower-bound of [6] (see Section 2). Whether this can be reached by PMD methods remains open, though using a more suitable Q-estimator than (10) with our step size regime and analysis, which extends to arbitrary Q-estimators, could bring the sample complexity closer to this.

# 6 Analysis

In this section, we present the proof of Theorem 4.1. A key component in establishing the $\gamma$-rate is avoiding the performance difference lemma that we state in Appendix B. In prior works, the quantity

that we are looking to bound $V^\star(\rho) - V^k(\rho)$ arises through the performance difference lemma. In particular, [38] use the lemma on $\mathbb{E}_{s \sim d_\rho^\star}[\langle Q_s^k, \pi_s^\star - \pi_s^{k+1} \rangle]$, which introduces a distribution mismatch coefficient in order to get a recursion. On the other hand, we extract the value sub-optimalities $V^\star(s) - V^k(s)$ and $\|V^\star - V^{k+1}\|_\infty$ directly from $\langle Q_s^k, \pi_s^\star - \pi_s^{k+1} \rangle$ in (12). This leads to an elegant analysis that may be of interest in the study of other methods, and ultimately allows us to remove distribution mismatch factors and obtain an exact $\gamma$-rate.

**Proof of Theorem 4.1:** Fix $s \in \mathcal{S}$ and $k \geq 0$. From Lemma A.2, we have that $\langle Q_s^k, \pi_s^{k+1} \rangle \leq \langle Q_s^{k+1}, \pi_s^{k+1} \rangle = V^{k+1}(s)$. This decouples the dependencies on $\pi^k$ and $\pi^{k+1}$ below and is one of the ingredients that allows us to bypass the performance difference lemma. Using this,

$$\langle Q_s^k, \pi_s^\star - \pi_s^{k+1} \rangle \geq \langle Q_s^k, \pi_s^\star \rangle - V^{k+1}(s) = \langle Q_s^k - Q_s^\star, \pi_s^\star \rangle + \langle Q_s^\star, \pi_s^\star \rangle - V^{k+1}(s)$$

$$\geq -\|Q_s^\star - Q_s^k\|_\infty + V^\star(s) - V^{k+1}(s), \qquad (11)$$

where the last step uses Hölder's inequality. Now we use that the difference in state-action values of different policies for the same state-action pair propagates the error to the next time-step, which is discounted by a factor of $\gamma$. Formally, for any state-action pair $(s, a) \in \mathcal{S} \times \mathcal{A}$,

$$Q^\star(s, a) - Q^k(s, a) = \gamma \sum_{s'} p(s'|s, a)(V^\star(s') - V^k(s'))$$

$$\leq \gamma \sum_{s'} p(s'|s, a)\|V^\star - V^k\|_\infty = \gamma\|V^\star - V^k\|_\infty,$$

which is the same phenomenon that is responsible for the contraction of the Bellman operator. This gives $\|Q_s^\star - Q_s^k\|_\infty \leq \gamma\|V^\star - V^k\|_\infty$. Plugging into Equation (11),

$$V^\star(s) - V^{k+1}(s) - \gamma\|V^\star - V^k\|_\infty \leq \langle Q_s^k, \pi_s^\star - \pi_s^{k+1} \rangle.$$

The rest of the proof relies on making the right-hand side of the above arbitrarily small by taking a large enough step size. Choose any greedy policy with respect to $Q_s^k$, $\widetilde{\pi}_s^{k+1} \in \widetilde{\Pi}_s^{k+1}$,

$$V^\star(s) - V^{k+1}(s) - \gamma\|V^\star - V^k\|_\infty \leq \langle Q_s^k, \pi_s^\star - \pi_s^{k+1} \rangle \qquad (12)$$

$$\leq \langle Q_s^k, \widetilde{\pi}_s^{k+1} - \pi_s^{k+1} \rangle \qquad (13)$$

where we use that $\widetilde{\pi}_s^{k+1}$ is greedy with respect to $Q_s^k$. We then apply Lemma A.1 or (16) to $p = \widetilde{\pi}_s^{k+1}$,

$$\langle Q_s^k, \widetilde{\pi}_s^{k+1} - \pi_s^{k+1} \rangle \leq \frac{D(\widetilde{\pi}_s^{k+1}, \pi_s^k) - D(\widetilde{\pi}_s^{k+1}, \pi_s^{k+1}) - D(\pi_s^{k+1}, \pi_s^k)}{\eta_k} \leq D(\widetilde{\pi}_s^{k+1}, \pi_s^k)/\eta_k.$$

Combining with (13) and noting that this holds for any $\widetilde{\pi}_s^{k+1} \in \widetilde{\Pi}_s^{k+1}$, we have

$$V^\star(s) - V^{k+1}(s) - \gamma\|V^\star - V^k\|_\infty \leq \frac{1}{\eta_k}\min_{\widetilde{\pi}_s^{k+1} \in \widetilde{\Pi}_s^{k+1}} D(\widetilde{\pi}_s^{k+1}, \pi_s^k) \leq c_k$$

from the step-size condition in the statement of the theorem. Rearranging and recalling that $s$ and $k$ were arbitrary, we can choose $s$ where $V^\star(s) - V^{k+1}(s)$ reaches its maximum value. We get

$$\|V^\star - V^{k+1}\|_\infty \leq \gamma\|V^\star - V^k\|_\infty + c_k,$$

and unravelling this recursion completes the proof. ∎

## 7  Conclusion

In this paper, we have shown that the general family of exact policy mirror descent algorithms in tabular MDPs under an adaptive step-size achieve the same dimension-free linear $\gamma$-rate of convergence of classical algorithms such as policy iteration. We provide matching lower-bounds that establish this rate as optimal for PMD and the adaptive step-size as necessary. We exploit a new approach to study the convergence of PMD, for which avoiding the performance difference lemma is a key element. Though the focus of our work is on the exact policy evaluation setting, the analysis naturally extends to the inexact setting, given access to an estimator of the action-value of a policy. We provide a result for a simple estimator under a generative model that improves upon the best-known sample complexity, although it still does not match the lower bound. Our method is general and applies to any estimator, meaning our result could be improved by a better estimator. Exploiting further algorithmic properties of PMD in the inexact setting may be needed to bridge the gap to the optimal sample complexity, and determine if PMD can match the lower bound in the inexact setting.

## Acknowledgments and Disclosure of Funding

EJ is funded by EPSRC through the Modern Statistics and Statistical Machine Learning (StatML) CDT (grant no. EP/S023151/1).

We would like to thank the reviewers and meta-reviewers for their time and feedback, which led to a better version of this paper.

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

# A  Properties of PMD

We present lemmas relevant to the analysis of PMD. Key to the analysis is the Three-Point Descent Lemma, that relates the improvement of the proximal gradient update compared to an arbitrary point. It originally comes from [14] (Lemma 3.2) where a proof can be found, though we use a slightly modified version from [38] (Lemma 6).

**Lemma A.1** (Three-Point Descent Lemma, Lemma 6 in [38]). *Suppose that $\mathcal{C} \subset \mathbb{R}^n$ is a closed convex set, $\phi : \mathcal{C} \to \mathbb{R}$ is a proper, closed convex function, $D_h(\cdot, \cdot)$ is the Bregman divergence generated by a function $h$ of Legendre type and rint domh $\cap\, \mathcal{C} \neq \emptyset$. For any $x \in$ rint domh, let*

$$x^+ = argmin_{u \in C}\{\phi(u) + D_h(u, x)\}. \tag{14}$$

*Then $x^+ \in$ rint dom $h \cap C$ and $\forall u \in C$,*

$$\phi(x^+) + D_h(x^+, x) \leq \phi(u) + D_h(u, x) - D_h(u, x^+) \tag{15}$$

The update (4) of PMD is an instance of the proximal minimisation (14) with $\mathcal{C} = \Delta(\mathcal{A})$, $x = \pi_s^k$ and $\phi(x) = -\eta_k \langle Q_s^k, x \rangle$. Plugging these into (15), Lemma A.1 relates the decrease in the proximal objective of $\pi_s^{k+1}$ to any other policy, i.e. $\forall p \in \Delta(\mathcal{A})$,

$$-\eta_k \langle Q_s^k, \pi_s^{k+1} \rangle + D_h(\pi_s^{k+1}, \pi_s^k) \leq -\eta_k \langle Q_s^k, p \rangle + D_h(p, \pi_s^k) - D_h(p, \pi_s^{k+1}). \tag{16}$$

This equation is key to the analysis in Section 6. In particular, it allows us to prove the following lemma regarding the monotonic improvement in action-value of PMD iterates. This is an extension of Lemma 7 in [38].

**Lemma A.2.** *Consider the policies produced by the iterative updates of PMD in (4). Then for any $k \geq 0$,*

$$Q^{k+1}(s, a) \geq Q^k(s, a), \quad \forall (s, a) \in \mathcal{S} \times \mathcal{A}.$$

## A.1  Proof of Lemma A.2

We first present Lemma 7 from [38], from which Lemma A.2 almost immediately follows.

**Lemma A.3** (Descent Property of PMD, Lemma 7 in [38]). *Consider the policies produced by the iterative updates of PMD in (4). Then for any $k \geq 0$*

$$\langle Q_s^k, \pi_s^{k+1} - \pi_s^k \rangle \geq 0, \quad \forall s \in \mathcal{S},$$

$$V^{k+1}(\rho) \geq V^k(\rho), \quad \forall \rho \in \Delta(\mathcal{S}).$$

*Proof.* From [38]. Recall that the Three-Point Descent Lemma states that $\forall p \in \Delta(\mathcal{A})$,

$$-\eta_k \langle Q_s^k, \pi_s^{k+1} \rangle + D_h(\pi_s^{k+1}, \pi_s^k) \leq -\eta_k \langle Q_s^k, p \rangle + D_h(p, \pi_s^k) - D_h(p, \pi_s^{k+1}).$$

Using this with $p = \pi_s^k$,

$$D_h(\pi_s^k, \pi_s^{k+1}) + D_h(\pi_s^{k+1}, \pi_s^k) \leq \eta_k \langle Q_s^k, \pi_s^{k+1} - \pi_s^k \rangle$$

and since the Bregman divergences are none-negative and $\eta_k > 0$,

$$0 \leq \langle Q_s^k, \pi_s^{k+1} - \pi_s^k \rangle$$

and the result follows by an application of the performance difference lemma (Appendix B)

$$V^{k+1}(\rho) - V^k(\rho) = \frac{1}{1 - \gamma} \mathbb{E}_{s \sim d_\rho^{k+1}} \left[ \langle Q_s^k, \pi_s^{k+1} - \pi_s^k \rangle \right]$$

$$\geq 0.$$

Note that we use the performance difference lemma here because it gives a simple concise proof, but we do not actually need to. To maintain our claim that we avoid the use of the performance difference lemma, we can get the same result without it. We sketch how to do this as follows. From the first part of the lemma, we have

$$\langle Q_s^k, \pi_s^{k+1} \rangle \geq \langle Q_s^k, \pi_s^k \rangle = V^k(s),$$

in all states $s$. Now note that the left hand side above is

$$\langle Q_s^k, \pi_s^{k+1} \rangle = \sum_a \pi^{k+1}(a|s) Q^k(s,a)$$

$$= \sum_a \pi^{k+1}(a|s) \Big( r(s,a) + \gamma \sum_{s'} p(s'|s,a) V^k(s') \Big)$$

and we can then apply $\langle Q_{s'}^k, \pi_{s'}^{k+1} \rangle \geq V^k(s')$ at state $s'$:

$$V^k(s) \leq \sum_a \pi^{k+1}(a|s) \Big( r(s,a) + \gamma \sum_{s'} p(s'|s,a) V^k(s') \Big)$$

$$\leq \sum_a \pi^{k+1}(a|s) \Big( r(s,a) + \gamma \sum_{s'} p(s'|s,a) \sum_{a'} \pi^{k+1}(a'|s') \Big( r(s',a') + \gamma \sum_{s''} p(s''|s',a') V^k(s'') \Big) \Big)$$

and proceeding iteratively in the limit you get exactly $V^{k+1}(s)$. ■

Since Lemma A.3 holds for any $\rho \in \Delta(\mathcal{S})$, it guarantees that the value in each state is non-decreasing for an update of PMD, i.e for all $s \in \mathcal{S}$,

$$V^{k+1}(s) - V^k(s) \geq 0.$$

Using this, we get

$$Q^{k+1}(s,a) - Q^k(s,a) = \gamma \sum_{s' \in \mathcal{S}} p(s'|s,a) \Big( V^{k+1}(s') - V^k(s') \Big) \geq 0,$$

which concludes the proof. ■

### A.2 Extension of Lemma A.2 to inexact setting:

As in the exact case, we first present Lemma 12 from [38] which is the extension of Lemma A.3 to the inexact case. We note that in the inexact case, we lose the monotonic increase of values due to the inaccuracy in our estimate $\widehat{Q}^k$ of $Q_s^k$.

**Lemma A.4.** *Consider the policies produced by the iterative updates of IPMD in (9). For any $k \geq 0$, if $\|\widehat{Q}_s^k - Q_s^k\|_\infty \leq \tau$, then*

$$\langle \widehat{Q}_s^k, \pi_s^{k+1} - \pi_s^k \rangle \geq 0, \quad \forall s \in \mathcal{S},$$

$$V^{k+1}(\rho) \geq V^k(\rho) - \frac{2\tau}{1-\gamma}, \quad \forall \rho \in \Delta(\mathcal{S}).$$

*Proof.* From [38]. The Three-Point Descent Lemma applied to the IPMD update (9) gives $\forall p \in \Delta(\mathcal{A})$,

$$-\eta_k \langle \widehat{Q}_s^k, \pi_s^{k+1} \rangle + D_h(\pi_s^{k+1}, \pi_s^k) \leq -\eta_k \langle \widehat{Q}_s^k, p \rangle + D_h(p, \pi_s^k) - D_h(p, \pi_s^{k+1}).$$

Using this with $p = \pi_s^k$,

$$D_h(\pi_s^k, \pi_s^{k+1}) + D_h(\pi_s^{k+1}, \pi_s^k) \leq \eta_k \langle \widehat{Q}_s^k, \pi_s^{k+1} - \pi_s^k \rangle$$

and since the Bregman divergences are none-negative and $\eta_k > 0$,

$$0 \leq \langle \widehat{Q}_s^k, \pi_s^{k+1} - \pi_s^k \rangle,$$

which proves the first inequality. Now we cannot use the above inequality directly with the performance difference lemma since $\widehat{Q}_s^k$ is not the true action-value. Instead, we have

$$V^{k+1}(\rho) - V^k(\rho) = \frac{1}{1-\gamma} \mathbb{E}_{s \sim d_\rho^{k+1}} \Big[ \langle Q_s^k, \pi_s^{k+1} - \pi_s^k \rangle \Big]$$

$$= \frac{1}{1-\gamma} \mathbb{E}_{s \sim d_\rho^{k+1}} \Big[ \langle Q_s^k - \widehat{Q}_s^k, \pi_s^{k+1} - \pi_s^k \rangle + \langle \widehat{Q}_s^k, \pi_s^{k+1} - \pi_s^k \rangle \Big]$$

$$\geq \frac{1}{1-\gamma} \mathbb{E}_{s \sim d_\rho^{k+1}} \Big[ -\|Q_s^k - \widehat{Q}_s^k\|_\infty \|\pi_s^{k+1} - \pi_s^k\|_1 \Big]$$

$$\geq \frac{1}{1-\gamma} \mathbb{E}_{s \sim d_\rho^{k+1}} \Big[ -2\tau \Big]$$

$$= -\frac{2\tau}{1-\gamma}$$

which concludes the proof. ■

Using the above lemma, we can state and prove the extension of Lemma A.2 to the inexact setting.

**Lemma A.5.** *Consider the policies produced by the iterative updates of IPMD in (9). For any $k \geq 0$, if $\|\widehat{Q}_s^k - Q_s^k\|_\infty \leq \tau$, then*

$$\widehat{Q}^{k+1}(s,a) \geq \widehat{Q}^k(s,a) - \frac{2\tau\gamma}{1-\gamma}, \quad \forall (s,a) \in \mathcal{S} \times \mathcal{A}.$$

*Proof.* As in the exact case, since Lemma A.4 holds for any $\rho \in \Delta(\mathcal{S})$, it applies to each state, i.e for all $s \in \mathcal{S}$,

$$V^{k+1}(s) - V^k(s) \geq -\frac{2\tau}{1-\gamma}.$$

Using this, we immediately have

$$Q^{k+1}(s,a) - Q^k(s,a) = \gamma \sum_{s' \in \mathcal{S}} p(s'|s,a)\Big(V^{k+1}(s') - V^k(s')\Big) \geq \frac{-2\tau\gamma}{1-\gamma},$$

which concludes the proof. ■

# B   Performance difference lemma

**Lemma B.1** (Performance Difference Lemma). *For any $\pi, \pi' \in \Pi$, we have*

$$V^\pi(\rho) - V^{\pi'}(\rho) = \frac{1}{1-\gamma}\mathbb{E}_{s \sim d_\rho^\pi}\Big[\langle Q_s^{\pi'}, \pi_s - \pi'_s \rangle\Big].$$

The performance difference lemma [19] is a property that relates the difference in values of policies to the policies themselves. The proof can be found in their paper under Lemma 6.1.

# C   Guarantees of Theorem 4.1 for various step-size choices

We give here two more choices of $\{c_k\}_{k \in \mathbb{Z}_{\geq 0}}$ for the step-size 5 of PMD and their corresponding guarantees from Theorem 4.1:

- $c_i = c_0$ for some $c_0 > 0$ yields a step-size with a constant component. The resulting bound is

$$\|V^\star - V^k\|_\infty \leq \gamma^k\|V^\star - V^0\|_\infty + \frac{c_0}{1-\gamma},$$

  which converges linearly up to some accuracy controlled by $c_0$.

- $c_i = \gamma^{i+1}c_0$ for some initial $c_0 > 0$ will yield a step-size with a component that is geometrically increasing as in [38], though at a slower rate than the one discussed in Section 4. The resulting bound is

$$\|V^\star - V^k\|_\infty \leq \gamma^k\Big(\|V^\star - V^0\|_\infty + kc_0\Big),$$

  which converges linearly with the sought-for $\gamma$-rate, though in early iterations the $k$ factor may dominate.

## D    Simulations

In this Section, we present simulations (see Figures 1-4 below) made by running NPG (PMD with negative-entropy mirror map, see [22]) on the hard MDP used to prove Theorem 4.2 - see Appendix E for the details of the construction. We use $n = 25$ so $|\mathcal{S}| = 51$ and $|\mathcal{A}| = 2$. We use $\delta = (1 - \gamma)\gamma^n/100$, $\gamma = 0.99$, $\pi^0(a_1|s) = \alpha$ (note that $a_1$ is the optimal action in each state). In each of the plots, we compare the performance of NPG using our step-size (5) from Theorem 4.1 (denoted **Adaptive** in the plots and by ADA in the discussion below) with $c_k = \gamma^{2k}$ and $c_0 = 1$ and the step-size of [38] $\eta_k = \eta_0/\gamma^k$ (denoted by INC in the discussion below) for a fixed $\eta_0 = 1$. In each of the plots, the left plot is $\|V^\star - V^k\|_\infty$ against iterations and the right plot is $\eta_k$ against iterations.

In all plots, the green curve represents the $\gamma$-rate. In particular it is the curve of $y = \gamma^x$.

**Discussion**

With fixed initial step-size $\eta_0 = 1$ for INC, the convergence of NPG with INC can be made arbitrarily slow in early iterations by choosing $\alpha$ $(= \pi^0(a_1|s))$ closer to 0 (Figures 1-3). The convergence of NPG with ADA is unaffected by small values of $\alpha$ because it adapts to them.

These observations agree with the theoretical results we established. Our adaptive step-size (5) achieves the $\gamma$-rate (Theorem 4.1) and this is optimal for iterations $k$ up until $n = 25$ (Theorem 4.2). The adaptivity in the step-size is necessary to achieve the $\gamma$-rate in these early iterations (Theorem 4.3).

In later iterations (Figure 1: iterations $\approx 100$. Figure 2: iterations $\approx 200$), INC has faster convergence than ADA. In these later iterations, acting greedily with respect to the action value of the current policy gives the optimal policy so INC converges faster because the step-size is larger (see Figure 1-2 right plot). This suggests combining ADA and INC by taking the maximum of both (Figure 4), yielding faster convergence than both. This is encouraging for future-work on super-linear convergence (as discussed in [38]) beyond the optimal $\gamma$-rate valid up to a finite iteration (see Section 4.1).

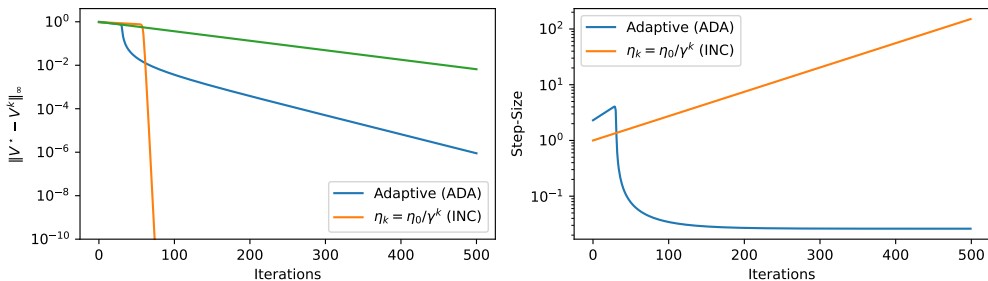

Figure 1: $\alpha = 10^{-1}$. Green curve: $y = \gamma^x$.

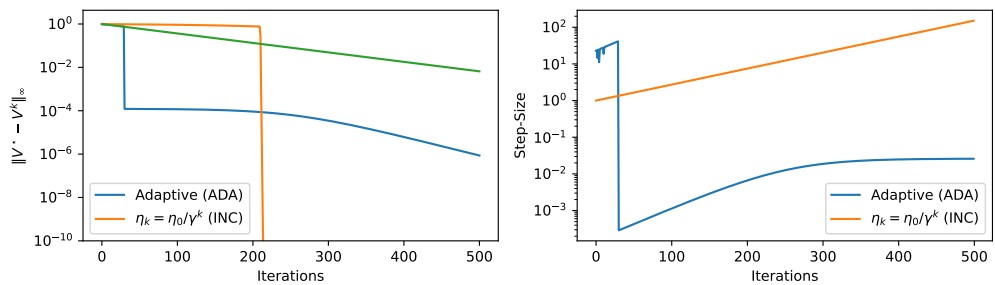

Figure 2: $\alpha = 10^{-10}$. Green curve: $y = \gamma^x$.

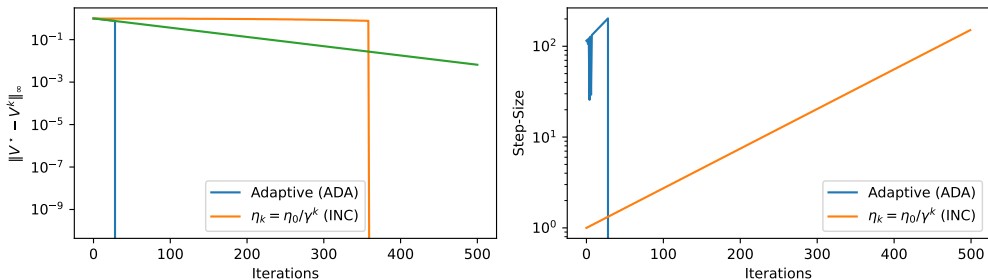

Figure 3: $\alpha = 10^{-50}$. Green curve: $y = \gamma^x$.

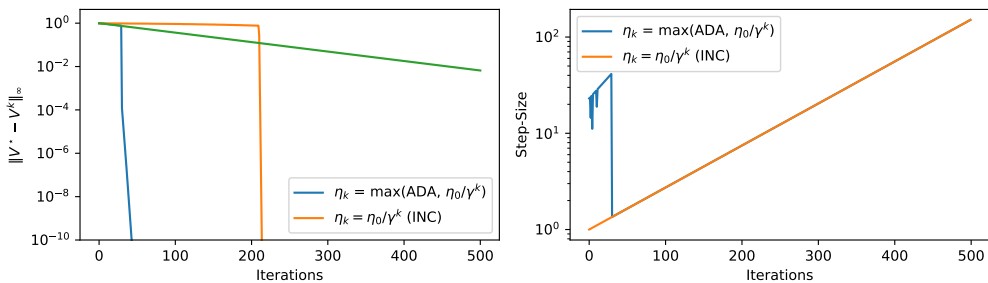

Figure 4: $\alpha = 10^{-10}$. $\eta_k = \max(\text{Adaptive}, \eta_0/\gamma^k)$ means the step-size is chosen as the largest between our "Adaptive" and the geometrically increasing of [38]. Green curve: $y = \gamma^x$.

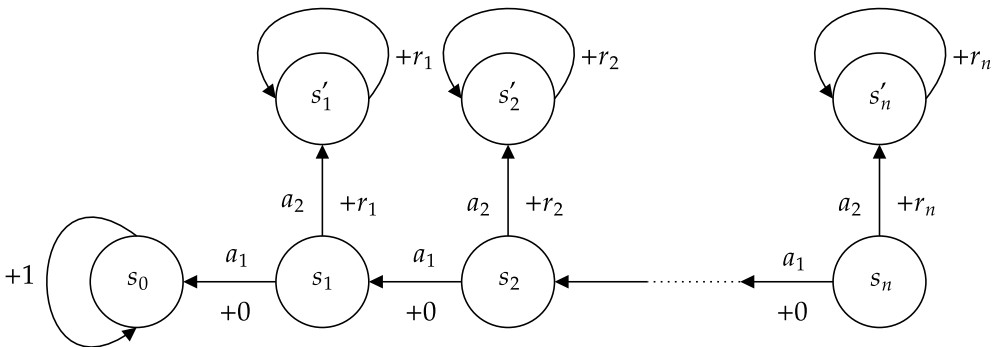

Figure 5: Example MDP used in the proof of Theorem 4.2

## E   Proof of Theorem 4.2

Fix $n > 0$ and $\delta \in (0, (1 - \gamma)\gamma^n)$. Consider the MDP shown in Figure 5. The state space is $\mathcal{S} = \{s_0, s_1, s_1', ..., s_n, s_n'\}$ and the action space is $\mathcal{A} = \{a_1, a_2\}$. There is a chain of states of length $n + 1$ with the states indexed from 0 to $n$. The left-most state $(s_0)$ is absorbing with reward +1. In the other states in the chain ($s_i$ for $i = 1, ..., n$), the agent can take action $a_1$ and move left (to $s_{i-1}$) with reward of 0, or take action $a_2$ and move to an additional absorbing state unique to the state it is currently in ($s_i'$) with reward $r_i = \gamma^{i+1} + \delta$ (that the agent also receives in that state for all future time-steps). Summarising, we have for $1 \leq i \leq n$

$$p(s_{i-1}|s_i, a_1) = 1, \quad r(s_i, a_1) = 0,$$
$$p(s_i'|s_i, a_2) = 1, \quad r(s_i, a_2) = r_i = \gamma^{i+1} + \delta,$$
$$p(s_i'|s_i', a) = 1, \quad r(s_i', a) = r_i = \gamma^{i+1} + \delta \quad \forall a \in \mathcal{A}.$$

The value of $\delta$ is carefully restricted so that the optimal action in all the states of the chain is $a_1$. The proof will consist in showing that if the agent starts with an initial policy that places most probability mass on the sub-optimal action $a_2$, then it has to learn that $a_1$ is the optimal action in the state directly to the left before it can start switching from action $a_2$ to $a_1$ in the current state. And this can at best happen one iteration at a time starting starting from the left-most state. In particular, we consider $\pi^0$ s.t $\pi^0(a_1|s) = \alpha$, $\pi^0(a_2|s) = 1 - \alpha$ for all states and some $\alpha$ s.t $0 < \alpha \leq \delta(1 - \gamma)$. We make the following claim from which the result will follow straightforwardly.

**Claim:** Fix $k < n$. The policies produced by PMD satisfy $\pi^k(a_1|s_i) \leq \alpha$ for $k < i \leq n$.

We prove this claim by induction.

**Base Case:** We want to show that $\pi^1(a_1|s_i) \leq \alpha$ for $i > 1$. We do this by showing that $Q^0(s_i, a_1) \leq Q^0(s_i, a_2)$ for $i > 1$ so that the probability of $\pi^1(a_1|s_i)$ cannot increase w.r.t $\pi^0(a_1|s_i)$, which is $\alpha$

(this follows from $\langle Q_s^k, \pi_s^{k+1} - \pi_s^k \rangle \geq 0$ for all iterations of PMD). We have:

$$
\begin{aligned}
Q^0(s_i, a_1) &= \gamma V^0(s_{i-1}) \\
&= \gamma \Big( \alpha Q^0(s_{i-1}, a_1) + (1-\alpha)Q^0(s_{i-1}, a_2) \Big) \\
&\leq \gamma \Big( \alpha \frac{\gamma^{i-1}}{1-\gamma} + \frac{r_{i-1}}{1-\gamma} \Big) \\
&\overset{(a)}{\leq} \gamma \Big( \delta(1-\gamma)\frac{\gamma^{i-1}}{1-\gamma} + \frac{\gamma^i + \delta}{1-\gamma} \Big) \\
&= \frac{\gamma^{i+1}}{1-\gamma} + \frac{\delta\gamma(1 + \gamma^{i-1} - \gamma^i)}{1-\gamma} \\
&\overset{(b)}{\leq} \frac{\gamma^{i+1}}{1-\gamma} + \frac{\delta}{1-\gamma} \\
&= Q^0(s_i, a_2),
\end{aligned}
$$

where we used $\alpha \leq \delta(1-\gamma)$ in (a) and $\gamma(1 + \gamma^{i-1} - \gamma^i) < 1$ for $\gamma \in [0,1)$ in (b). This concludes the base case.

**Inductive Step:** Now assume that the claim is true for $k$ and we want to show that $\pi^{k+1}(a_1|s_i) \leq \alpha$ for $i > k+1$. We do this in the same way as the base case by showing that $Q^k(s_i, a_1) \leq Q^k(s_i, a_2)$ for $i > k+1$ so that the probability of $\pi^{k+1}(a_1|s_i)$ cannot increase w.r.t $\pi^k(a_1|s_i)$, which is less than or equal to $\alpha$ by the inductive hypothesis. We have:

$$
\begin{aligned}
Q^k(s_i, a_1) &= \gamma V^k(s_{i-1}) \\
&= \gamma \Big( \pi^k(a_1|s_{i-1})Q^k(s_{i-1}, a_1) + \pi^k(a_2|s_{i-1})Q^k(s_{i-1}, a_2) \Big) \\
&\overset{(a)}{\leq} \gamma \Big( \alpha Q^k(s_{i-1}, a_1) + Q^k(s_{i-1}, a_2) \Big) \\
&\leq \gamma \Big( \alpha \frac{\gamma^{i-1}}{1-\gamma} + \frac{r_{i-1}}{1-\gamma} \Big) \\
&\overset{(b)}{\leq} \gamma \Big( \delta(1-\gamma)\frac{\gamma^{i-1}}{1-\gamma} + \frac{\gamma^i + \delta}{1-\gamma} \Big) \\
&= \frac{\gamma^{i+1}}{1-\gamma} + \frac{\delta\gamma(1 + \gamma^{i-1} - \gamma^i)}{1-\gamma} \\
&\overset{(c)}{\leq} \frac{\gamma^{i+1}}{1-\gamma} + \frac{\delta}{1-\gamma} \\
&= Q^k(s_i, a_2),
\end{aligned}
$$

where we used in (a) that $\pi^k(a_1|s_{i-1}) \leq \alpha$ for $i > k+1$, which is true by the inductive hypothesis since $i-1 > k$, in (b) that $\alpha \leq \delta(1-\gamma)$ and in (c) that $\gamma(1 + \gamma^{i-1} - \gamma^i) < 1$ for $\gamma \in [0,1)$. This concludes the proof of the claim.

Now using the claim

$$
\begin{aligned}
V^k(s_{k+1}) &= \pi^k(a_1|s_{k+1})Q^k(s_{k+1}, a_1) + \pi^k(a_2|s_{k+1})Q^k(s_{k+1}, a_2) \\
&\leq \alpha \frac{\gamma^{k+1}}{1-\gamma} + \frac{r_{k+1}}{1-\gamma} \\
&= \alpha \frac{\gamma^{k+1}}{1-\gamma} + \frac{\gamma^{k+2} + \delta}{1-\gamma},
\end{aligned}
$$

so

$$V^\star(s_{k+1}) - V^k(s_{k+1}) \geq \frac{\gamma^{k+1}}{1-\gamma} - \alpha \frac{\gamma^{k+1}}{1-\gamma} - \frac{\gamma^{k+2}+\delta}{1-\gamma}$$

$$= \frac{\gamma^{k+1}(1-\gamma)}{1-\gamma} - \frac{\alpha\gamma^{k+1}+\delta}{1-\gamma}$$

$$\geq \gamma^{k+1} - \frac{\alpha+\delta}{1-\gamma}$$

$$\geq \gamma^{k+1} - \frac{2\delta}{1-\gamma}, \tag{17}$$

where we used that $\alpha \leq \delta$. Now note that

$$V^0(s_1) = \alpha Q^0(s_1, a_1) + (1-\alpha)Q^0(s_1, a_2)$$

$$= \alpha \frac{\gamma}{1-\gamma} + (1-\alpha)\frac{\gamma^2+\delta}{1-\gamma},$$

so

$$V^\star(s_1) - V^0(s_1) = \frac{\gamma}{1-\gamma} - \alpha\frac{\gamma}{1-\gamma} - (1-\alpha)\frac{\gamma^2+\delta}{1-\gamma}$$

$$= (1-\alpha)\frac{\gamma}{1-\gamma} - (1-\alpha)\frac{\gamma^2+\delta}{1-\gamma}$$

$$= \frac{1-\alpha}{1-\gamma}\left(\gamma - \gamma^2 - \delta\right)$$

$$\leq \frac{1-\alpha}{1-\gamma}\left(\gamma - \gamma^2\right)$$

$$= \gamma\frac{1-\alpha}{1-\gamma}\left(1-\gamma\right)$$

$$= \gamma(1-\alpha)$$

$$\leq \gamma$$

and by induction we can show this is the case for all states (above is base case), the inductive step is as follows (assuming $V^\star(s_k) - V^0(s_k) \leq \gamma$),

$$V^\star(s_{k+1}) - V^0(s_{k+1}) = \frac{\gamma^{k+1}}{1-\gamma} - (1-\alpha)\frac{\gamma^{k+2}+\delta}{1-\gamma} - \alpha\gamma V^0(s_k)$$

$$= (1-\alpha)\left[\frac{\gamma^{k+1} - \gamma^{k+2} - \delta}{1-\gamma}\right] + \alpha\gamma\left[V^\star(s_k) - V^0(s_k)\right]$$

$$\leq (1-\alpha)\gamma^{k+1} + \alpha\gamma^2$$

$$\leq \gamma$$

and so

$$\|V^\star - V^0\|_\infty \leq \gamma,$$

which combining with (17) gives,

$$V^\star(s_{k+1}) - V^k(s_{k+1}) \geq \gamma^k \|V^\star - V^0\|_\infty - \frac{2\delta}{1-\gamma}$$

$$\implies \|V^\star - V^k\|_\infty \geq \gamma^k \|V^\star - V^0\|_\infty - \frac{2\delta}{1-\gamma}.$$

Choosing $\delta = \frac{(1-\gamma)}{4}\gamma^n\|V^\star - V^0\|_\infty \leq \frac{(1-\gamma)}{4}\gamma^{n+1}$ (which is in the valid range of $\delta$) concludes the proof since $\frac{2\delta}{1-\gamma} \leq \frac{1}{2}\gamma^k\|V^\star - V^0\|_\infty$. $\blacksquare$

# F Proof of Theorem 4.3

Consider the same MDP as in the proof of Theorem 4.2 in Appendix E (see Figure 5). Denote $c = \frac{1-\gamma}{8}$ and note that $c < \frac{\sqrt{\gamma}}{1+\sqrt{\gamma}}\frac{1-\gamma}{2}$ since $\frac{1}{4} < \frac{\sqrt{\gamma}}{1+\sqrt{\gamma}}$ for $\gamma > 0.2$.

Suppose you consider NPG updates with initial policy $\pi^0(a_1|s_i) = \alpha$. Recall that NPG is the instance of PMD with relative entropy as the mirror map. It can be shown that NPG has the closed form update

$$\pi^{k+1}(a|s) = \frac{\pi^k(a|s)e^{\eta_k Q^k(s,a)}}{\sum_{a'} \pi^k(a'|s)e^{\eta_k Q^k(s,a')}}.$$

We know from the proof of Theorem E that for any step-size regime, for $i > k+1$

$$Q^k(s_i, a_1) \leq Q^k(s_i, a_2).$$

Now, $\|V^\star - V^0\|_\infty = V^\star(s_1) - V^0(s_1) \leq \gamma - \frac{\delta}{1-\gamma}$ (see Section F.1 below). The idea of the proof is to show that satisfying the bound given in the statement of the theorem will imply that a certain condition on the step-size.

Fix a state $s_k$ and let $k_0$ be the first iteration where $Q^{k_0}(s_k, a_1) > Q^{k_0}(s_k, a_2)$. By the above, we must have $k \leq k_0 + 1$, or $k_0 \geq k - 1$. By the proof of Theorem E, we also have $\pi^{k_0}(a_1|s_k) \leq \alpha$ (before iteration $k_0$, $Q(s_k, \cdot)$ favors $a_2$, so $\pi^{k_0}(a_1|s_k)$ has not increased compared to $\pi^0(a_1|s_k) = \alpha$).

We want a $\gamma$-contraction at every iteration, i.e. we assume the following is satisfied:

$$V^\star(s_k) - V^{k_0+1}(s_k) \leq \gamma^{k_0+1}(\|V^\star - V^0\|_\infty + c) \leq \gamma^{k_0+1}(\gamma - \frac{\delta}{1-\gamma} + c).$$

Now, by direct computation,

$$V^\star(s_k) - V^{k_0+1}(s_k) = \pi^{k_0+1}(a_1|s_k)\gamma(V^\star(s_{k-1}) - V^{k_0+1}(s_{k-1})) + \pi^{k_0+1}(a_2|s_2)\frac{\gamma^k - r_k}{1-\gamma}$$

$$\geq \pi^{k_0+1}(a_2|s_2)\frac{\gamma^k - r_k}{1-\gamma} = \pi^{k_0+1}(a_2|s_2)(\gamma^k - \frac{\delta}{1-\gamma}).$$

Putting this together with the above (this is an implication as this is about the necessity rather than sufficiency), we must have:

$$\pi^{k_0+1}(a_2|s_2)(\gamma^k - \frac{\delta}{1-\gamma}) \leq \gamma^{k_0+1}(\gamma - \frac{\delta}{1-\gamma} + c)$$

$$\implies \pi^{k_0+1}(a_2|s_2) \leq \frac{\gamma^{k_0+1}(\gamma - \frac{\delta}{1-\gamma} + c)}{(\gamma^k - \frac{\delta}{1-\gamma})} = \beta$$

If we choose $\delta < \frac{1}{2}(1-\gamma)(1-\sqrt{\gamma})\gamma^k$ then $\beta < \sqrt{\gamma}$ and require

$$\pi^{k_0+1}(a_2|s_2) \leq \sqrt{\gamma}.$$

To see this, start from $\beta \leq \sqrt{\gamma}$, this is equivalent to

$$\frac{\gamma^{k_0+1}(\gamma - \frac{\delta}{1-\gamma} + c)}{(\gamma^k - \frac{\delta}{1-\gamma})} \leq \sqrt{\gamma}$$

$$\Longleftarrow \frac{\gamma^k(\gamma - \frac{\delta}{1-\gamma} + c)}{(\gamma^k - \frac{\delta}{1-\gamma})} \leq \sqrt{\gamma} \quad \text{since } k_0 + 1 \geq k$$

$$\Longleftrightarrow \gamma^k(\gamma - \frac{\delta}{1-\gamma} + c) \leq \sqrt{\gamma}(\gamma^k - \frac{\delta}{1-\gamma})$$

$$\Longleftarrow \gamma^k(\gamma + c) \leq \sqrt{\gamma}(\gamma^k - \frac{\delta}{1-\gamma})$$

$$\Longleftrightarrow \gamma^{k-\frac{1}{2}}(\gamma + c) \leq \gamma^k - \frac{\delta}{1-\gamma}$$

$$\Longleftrightarrow \frac{\delta}{1-\gamma} \leq \gamma^{k-\frac{1}{2}}(\sqrt{\gamma} - \gamma - c)$$

$$\Longleftarrow \frac{\delta}{1-\gamma} \leq \gamma^{k-\frac{1}{2}}(\sqrt{\gamma} - \gamma - \frac{\sqrt{\gamma}}{1+\sqrt{\gamma}}\frac{1-\gamma}{2}) \quad \text{since } -c > -\frac{\sqrt{\gamma}}{1+\sqrt{\gamma}}\frac{1-\gamma}{2}$$

$$\Longleftrightarrow \frac{\delta}{1-\gamma} \leq \gamma^{k-\frac{1}{2}}(\sqrt{\gamma} - \gamma - \frac{\sqrt{\gamma}}{2}(1-\sqrt{\gamma}))$$

$$\Longleftrightarrow \frac{\delta}{1-\gamma} \leq \gamma^{k-\frac{1}{2}}(\sqrt{\gamma} - \gamma - \frac{\sqrt{\gamma}}{2} + \frac{\gamma}{2})$$

$$\Longleftrightarrow \frac{\delta}{1-\gamma} \leq \gamma^{k-\frac{1}{2}}(\frac{\sqrt{\gamma}}{2} - \frac{\gamma}{2})$$

$$\Longleftrightarrow \delta \leq \frac{1}{2}\gamma^k(1 - \sqrt{\gamma})(1 - \gamma),$$

which is the condition for $\delta$ we imposed initially.

To achieve the above condition $\pi^{k_0+1}(a_2|s_2) \leq \sqrt{\gamma}$, recalling that $\pi^{k_0}(a_2|s_2) \geq 1 - \alpha$, $\eta_{k_0}$ has to satisfy

$$\eta_{k_0} \geq \frac{1}{Q^{k_0}(s_k, a_1) - Q^{k_0}(s_k, a_2)}\left[\log((1-\alpha)(1-\sqrt{\gamma})) + KL(\tilde{\pi}_{s_k}^{k_0+1}, \pi_{s_k}^{k_0})\right]$$

To see this, again start from $\pi^{k_0+1}(a_2|s_2) \leq \sqrt{\gamma}$, this is equivalent to (use $k_0 = m$ for simplicity of notation) using the closed-form update of NPG:

$$\pi^m(a_2|s_2)\exp(\eta_m Q^m(s_k, a_2)) \leq$$
$$\sqrt{\gamma}(\pi^m(a_2|s_2)\exp(\eta_m Q^m(s_k, a_2)) + \pi^m(a_1|s_2)\exp(\eta_m Q^m(s_k, a_1)))$$

$$\Longleftrightarrow \frac{1}{\sqrt{\gamma}} \leq 1 + \frac{\pi^m(a_1|s_2)}{\pi^m(a_2|s_2)}\exp(\eta_m(Q^m(s_k, a_1) - Q^m(s_k, a_2)))$$

$$\Longleftrightarrow \frac{1-\sqrt{\gamma}}{\sqrt{\gamma}}\frac{\pi^m(a_2|s_2)}{\pi^m(a_1|s_2)} \leq \exp(\eta_m(Q^m(s_k, a_1) - Q^m(s_k, a_2)))$$

$$\Longleftrightarrow \eta_m(Q^m(s_k, a_1) - Q^m(s_k, a_2)) \geq \log\left(\frac{1-\sqrt{\gamma}}{\sqrt{\gamma}}\frac{\pi^m(a_2|s_2)}{\pi^m(a_1|s_2)}\right)$$

$$\Longleftrightarrow \eta_m \geq \frac{1}{Q^m(s_k, a_1) - Q^m(s_k, a_2)}\left[\log\left(\frac{1-\sqrt{\gamma}}{\sqrt{\gamma}}\pi^m(a_2|s_2)\right) + \log\left(\frac{1}{\pi^m(a_1|s_2)}\right)\right]$$

$$\Longrightarrow \eta_m \geq \frac{1}{Q^m(s_k, a_1) - Q^m(s_k, a_2)}\left[\log\left((1-\alpha)\frac{1-\sqrt{\gamma}}{\sqrt{\gamma}}\right) + KL(\tilde{\pi}_{s_k}^{m+1}, \pi_{s_k}^m)\right]$$

$$\Longrightarrow \eta_m \geq \frac{1}{Q^m(s_k, a_1) - Q^m(s_k, a_2)}\left[\log\left((1-\alpha)(1-\sqrt{\gamma})\right) + KL(\tilde{\pi}_{s_k}^{m+1}, \pi_{s_k}^m)\right].$$

As we take $\alpha \to 0$, the KL term will dominate. In particular, note $\alpha < 1 - \gamma$ so $1 - \alpha > \gamma$ and

$$(1-\alpha)(1-\sqrt{\gamma}) > \gamma(1-\sqrt{\gamma})$$

and if we further impose the condition $\alpha < \gamma^2(1 - \sqrt{\gamma})^2$ then

$$(1 - \alpha)(1 - \sqrt{\gamma}) > \sqrt{\alpha} > \sqrt{\pi^{k_0}(a_1|s_2)}$$

and the step-size needs to satisfy the following condition:

$$
\begin{aligned}
\eta_{k_0} &\geq \frac{1}{Q^{k_0}(s_k, a_1) - Q^{k_0}(s_k, a_2)}\left[\log(\sqrt{\pi^{k_0}(a_1|s_2)}) + KL(\tilde{\pi}_{s_k}^{k_0+1}, \pi_{s_k}^{k_0})\right] \\
&= \frac{1}{Q^{k_0}(s_k, a_1) - Q^{k_0}(s_k, a_2)}\left[-\frac{1}{2}KL(\tilde{\pi}_{s_k}^{k_0+1}, \pi_{s_k}^{k_0}) + KL(\tilde{\pi}_{s_k}^{k_0+1}, \pi_{s_k}^{k_0})\right] \\
&= \frac{1}{2(Q^{k_0}(s_k, a_1) - Q^{k_0}(s_k, a_2))}KL(\tilde{\pi}_{s_k}^{k_0+1}, \pi_{s_k}^{k_0})
\end{aligned}
\tag{18}
$$

**Distinct Iterations:** Note that the iteration $k_0(s_k)$ where $Q(\cdot, s_k)$ starts becoming bigger at $a_1$ that $a_2$ is distinct for each $s_k$. Fix any $s_k$ and $k_0 = k_0(s_k)$. We have

$$Q^{k_0}(s_k, a_1) < Q^{k_0}(s_k, a_2)$$
$$Q^{k_0+1}(s_k, a_2) \leq Q^{k_0+1}(s_k, a_1)$$

then $\pi^{k_0+1}(a_1|s_k) \leq \pi^{k_0}(a_1|s_k) \leq \alpha$ (since $Q^t$ points towards $a_2$ in $s_k$ for all $t \leq k_0$). Then applying exactly the same steps as in the proof of Theorem 4.2, we have

$$Q^{k_0+1}(s_{k+1}, a_1) < Q^{k_0+1}(s_{k+1}, a_2),$$

meaning that $k_0(s_k)$ is disctinct to $k_0(s_{k+1})$.

**Upper Bounding Q-value difference:** We want to upper-bound the Q-value difference appearing in the step-size condition above. We have,

$$Q^{k_0}(s_k, a_2) = \frac{r_k}{1 - \gamma} = \frac{\gamma^{k+1} + \delta}{1 - \gamma}$$
$$Q^{k_0}(s_k, a_1) = \gamma V^{k_0}(s_{k-1}) \leq \frac{\gamma^k}{1 - \gamma}.$$

So,

$$
\begin{aligned}
Q^{k_0}(s_k, a_1) - Q^{k_0}(s_k, a_2) &\leq \frac{\gamma^k}{1 - \gamma} - \frac{\gamma^{k+1} + \delta}{1 - \gamma} \\
&= \gamma^k - \frac{\delta}{1 - \gamma} \\
&\leq \gamma^k.
\end{aligned}
$$

Plugging this into the above bound (18), if the iterates of NPG are to satisfy the bound with the $\gamma$-rate in the statement of the theorem, the step-size must at least satisfy the following condition:

$$\eta_{k_0} \geq \frac{1}{2\gamma^k}KL(\tilde{\pi}_{s_k}^{k_0+1}, \pi_{s_k}^{k_0}),$$

which concludes the proof. ∎

### F.1 Largest sub-optimality gap at iteration 0

In this section, we prove the claim that

$$\|V^\star - V^0\|_\infty = V^\star(s_1) - V^0(s_1) \leq \gamma - \frac{\delta}{1 - \gamma}$$

**Proof:** First of all, $V^\star(s_1) - V^0(s_1) = \pi^0(a_2|s_1)\frac{\gamma - r_1}{1 - \gamma} = (1 - \alpha)(\gamma - \frac{\delta}{1-\gamma}) \leq \gamma - \frac{\delta}{1-\gamma}$. For the first part, we proceed by induction. We will use throughout that

$$\frac{\gamma^k - r_k}{1 - \gamma} = \gamma^k - \frac{\delta}{1 - \gamma} \leq V^\star(s_1) - V^0(s_1) = (1 - \alpha)(\gamma - \frac{\delta}{1 - \gamma}).$$

This is true if (when LHS is the largest)

$$\gamma^2 - \frac{\delta}{1-\gamma} \le (1-\alpha)(\gamma - \frac{\delta}{1-\gamma})$$

which holds when

$$\alpha \le \frac{\gamma(1-\gamma)^2}{\gamma(1-\gamma) - \delta}$$
$$\Longleftarrow \alpha \le 1 - \gamma$$

**Base Case:**

$$V^\star(s_2) - V^0(s_2) = \alpha\gamma(V^\star(s_1) - V^0(s_1)) + (1-\alpha)\frac{\gamma^2 - r_2}{1-\gamma}$$
$$\le \alpha\gamma(V^\star(s_1) - V^0(s_1)) + (1-\alpha)(V^\star(s_1) - V^0(s_1))$$
$$\le V^\star(s_1) - V^0(s_1)$$

**Inductive Step:** Assume true for $k$. Then,

$$V^\star(s_{k+1}) - V^0(s_{k+1}) = \alpha\gamma(V^\star(s_k) - V^0(s_k)) + (1-\alpha)\frac{\gamma^{k+1} - r_{k+1}}{1-\gamma}$$
$$\le \alpha\gamma(V^\star(s_1) - V^0(s_1)) + (1-\alpha)(V^\star(s_1) - V^0(s_1))$$
$$\le V^\star(s_1) - V^0(s_1),$$

which concludes the proof. ∎

# G    Inexact policy mirror descent and the generative model

The following Lemma from [38] controls the accuracy of the estimator $\widehat{Q}_s^k$ specified in (10) of Section 5 with respect to $H$ and $M_k$:

**Lemma G.1** (Lemma 15 in [38]). *Consider using (10) to estimate $Q_s^k$ for all state-action pairs for $K$ iterations of IPMD. Then for $\delta \in (0, 1)$, if for all $k \leq K$,*

$$M_k \geq \frac{\gamma^{-2H}}{2} log\big(\frac{2K|\mathcal{S}||\mathcal{A}|}{\delta}\big).$$

*Then with probability at least $1 - \delta$, we have for all $k \leq K$,*

$$\|\widehat{Q}_s^k - Q_s^k\|_\infty \leq \frac{2\gamma^H}{1 - \gamma}.$$

The proof of this result can be found in Lemma 15 of [38].

## G.1    Proof of Theorem 5.1

This proof is similar to that of [38] (Theorem 14). It is also similar in structure to the proof of Theorem 4.1 in Section 6.

Fix a state $s \in \mathcal{S}$ and an integer $k \geq 0$. For now let's assume that our Q-estimates are $\tau$-accurate ($\tau > 0$), i.e.

$$\|Q^k - \widehat{Q}^k\|_\infty \leq \tau$$

for all $k \geq 0$. With this assumption, we have from Lemma A.5 in Appendix A.1,

$$Q^{k+1}(s, a) \geq Q^k(s, a) - \frac{2\gamma\tau}{1 - \gamma}, \quad \forall(s, a) \in \mathcal{S} \times \mathcal{A}.$$

Now proceeding in a similar way to Section 6,

$$\begin{aligned}
\langle\widehat{Q}_s^k, \pi_s^\star - \pi_s^{k+1}\rangle &= \langle Q_s^k, \pi_s^\star - \pi_s^{k+1}\rangle + \langle\widehat{Q}_s^k - Q_s^k, \pi_s^\star - \pi_s^{k+1}\rangle \\
&\geq \langle Q_s^k, \pi_s^\star\rangle - \langle Q_s^k, \pi_s^{k+1}\rangle - \|\widehat{Q}_s^k - Q_s^k\|_\infty\|\pi_s^\star - \pi_s^{k+1}\|_1 \\
&\geq \langle Q_s^k, \pi_s^\star\rangle - \langle Q_s^{k+1}, \pi_s^{k+1}\rangle - \frac{2\gamma\tau}{1 - \gamma} - 2\tau \\
&\geq \langle Q_s^k, \pi_s^\star\rangle - V^{k+1}(s) - \frac{4\gamma\tau}{1 - \gamma} \\
&= \langle Q_s^k - Q_s^\star, \pi_s^\star\rangle + V^\star(s) - V^{k+1}(s) - \frac{4\gamma\tau}{1 - \gamma} \\
&\geq -\|Q_s^\star - Q_s^k\|_\infty + V^\star(s) - V^{k+1}(s) - \frac{4\gamma\tau}{1 - \gamma} \\
&\geq -\gamma\|V^\star - V^k\|_\infty + V^\star(s) - V^{k+1}(s) - \frac{4\gamma\tau}{1 - \gamma}.
\end{aligned}$$

Now again proceeding exactly as in Section 6 with this extra $\tau$-term using the step-size condition ($c_k = \gamma^{2k+1}$), we end up with

$$\|V^\star - V^{k+1}\|_\infty \leq \gamma\|V^\star - V^k\|_\infty + \gamma^{2k+1} + \frac{4\gamma\tau}{1 - \gamma}.$$

Unravelling this recursion yields

$$\begin{aligned}
\|V^\star - V^k\|_\infty &\leq \gamma^k\Big(\|V^\star - V^0\|_\infty + \sum_{i=1}^k \gamma^{-i}\gamma^{2(i-1)+1}\Big) + \frac{4\gamma\tau}{1 - \gamma}\sum_{i=0}^{k-1}\gamma^i \\
&\leq \gamma^k\Big(\|V^\star - V^0\|_\infty + \frac{1}{1 - \gamma}\Big) + \frac{4\gamma\tau}{(1 - \gamma)^2}.
\end{aligned}$$

Now using the properties of the estimator (10) in Lemma G.1, we have with probability $1 - \delta$ for all $0 \leq k \leq K$,

$$\tau = \frac{2\gamma^H}{1 - \gamma},$$

giving

$$\|V^\star - V^k\|_\infty \leq \gamma^k \Big( \|V^\star - V^0\|_\infty + \frac{1}{1 - \gamma} \Big) + \frac{8\gamma^H}{(1 - \gamma)^3}$$

$$\leq \frac{2}{1 - \gamma} \gamma^k + \frac{8\gamma^H}{(1 - \gamma)^3}.$$

This establishes the first bound. Now

$$K > \frac{1}{1 - \gamma} \log \frac{4}{(1 - \gamma)\varepsilon} \implies \frac{2}{1 - \gamma} \gamma^k \leq \varepsilon/2,$$

$$H \geq \frac{1}{1 - \gamma} \log \frac{16}{(1 - \gamma)^3 \varepsilon} \implies \frac{8\gamma^H}{(1 - \gamma)^3} \leq \varepsilon/2$$

giving

$$\|V^\star - V^k\|_\infty \leq \varepsilon/2 + \varepsilon/2 = \varepsilon$$

as required. In terms of M, we have

$$M \geq \frac{\gamma^{-2H}}{2} \log \frac{2K|\mathcal{S}||\mathcal{A}|}{\delta}$$

$$\geq \frac{1}{2} \Big( \frac{16}{(1 - \gamma)^3 \varepsilon} \Big)^2 \log \frac{2K|\mathcal{S}||\mathcal{A}|}{\delta}$$

$$= \frac{16^2}{2(1 - \gamma)^6 \varepsilon^2} \log \frac{2K|\mathcal{S}||\mathcal{A}|}{\delta}$$

and the corresponding number of calls to the sampling model, i.e. the sample complexity is (what we have shown above is actually a lower bound but can choose $K, H, M$ so that it is of the following order),

$$|\mathcal{S}| \cdot |\mathcal{A}| \cdot K \cdot H \cdot M = \tilde{O}\Big( \frac{|\mathcal{S}||\mathcal{A}|}{(1 - \gamma)^8 \varepsilon^2} \Big),$$

where the notation $\tilde{O}()$ hides poly-logarithmic factors. This completes the proof.  ∎

# H MDP examples

## H.1 MDP on which distribution-mismatch coefficient scales with size of state space

We construct an MDP on which

$$\theta_\rho = \frac{1}{1-\gamma} \left\| \frac{d_\rho^\star}{\rho} \right\|_\infty,$$

scales with $|\mathcal{S}|$, and hence so does the iteration complexity of the bound of [38] for exact PMD.

Consider an MDP with state-space $\mathcal{S} = \{s_1, s_2, ..., s_n\}$ of size n and arbitrary action space $\mathcal{A}$. $s_1$ is an absorbing state giving out rewards of 1 at each time-step, regardless of the action taken, i.e

$$p(s_1|s_1, a) = 1, \quad r(s_1, a) = 1 \quad \forall a \in \mathcal{A}.$$

All others states have an action, say $a_1$, that gives out a reward of 1 and with probability $1 - \delta$ brings the agent to state $s_1$ for some $\delta > 0$ and spreads the remaining $\delta$ probability arbitrarily amongst the other states. The other actions have arbitrary rewards strictly less than 1 associated to them, and arbitrary transition probabilities that place 0 mass on state $s_1$, i.e

$$p(s_1|s, a_1) = 1 - \delta, \quad r(s, a_1) = 1 \quad \forall s \neq s_1,$$
$$p(s_1|s, a) = 0, \quad r(s, a) < 1 \quad \forall s \neq s_1, \forall a \neq a_1.$$

Denote $r_{\max} = \max_{s \neq s_1, a \neq a_1} r(s, a) < 1$. The following condition ensures that $a_1$ is the optimal action in all states,

$$\delta \leq \frac{1-\gamma}{\gamma}(1 - r_{\max})$$

so that $\pi^\star(s) = a_1$ for all states $s$. To see this, consider $s_i \neq s_1$, $a_m \neq a_1$ and an arbitrary policy $\pi$,

$$Q^\pi(s_i, a_1) = 1 + \gamma\left(\frac{1-\delta}{1-\gamma} + \sum_{j=2}^n p(s_j|s_i, a_1)V^\pi(s_j)\right)$$

$$\geq 1 + \gamma\frac{1-\delta}{1-\gamma}$$

$$Q^\pi(s_i, a_m) = r(s_i, a_m) + \gamma\sum_{j=2}^n p(s_j|s_i, a_1)V^\pi(s_j)$$

$$\leq r_{\max} + \gamma\frac{1}{1-\gamma}$$

and solving

$$r_{\max} + \gamma\frac{1}{1-\gamma} \leq 1 + \gamma\frac{1-\delta}{1-\gamma}$$

will yield the condition above.

Then for $t \geq 1$ (abusing notation, $s_t$ denotes the state at time t),

$$\mathbb{P}^{\pi^\star}(s_t = s_1|s_0 = s) = \sum_{s'} \mathbb{P}^{\pi^\star}(s_t = s_1, s_{t-1} = s'|s_0 = s)$$

$$= \sum_{s'} p(s_1|s', a_1)\mathbb{P}^{\pi^\star}(s_{t-1} = s'|s_0 = s)$$

$$\geq \sum_{s'}(1-\delta)\mathbb{P}^{\pi^\star}(s_{t-1} = s'|s_0 = s)$$

$$= 1 - \delta$$

and

$$d_\rho^\star(s_1) = (1-\gamma)\sum_s \rho(s)\sum_{t=0}^\infty \gamma^t \mathbb{P}^{\pi^\star}(s_t = s_1|s_0 = s)$$

$$\geq (1-\gamma)\sum_s \rho(s)\sum_{t=1}^\infty \gamma^t(1-\delta)$$

$$\geq (1-\gamma)\sum_s \rho(s)\frac{\gamma}{1-\gamma}(1-\delta)$$

$$= \gamma(1-\delta).$$

Now

$$\left\|\frac{d_\rho^\star}{\rho}\right\|_\infty \geq \frac{d_\rho^\star(s_1)}{\rho(s_1)} \geq \frac{\gamma(1-\delta)}{\rho(s_1)}$$

and depending on what $\rho$ you consider, $\theta_\rho$ can be arbitrarily large. In particular, the natural choice of the uniform starting-state distribution $\rho(s) = 1/n$ leads to

$$\theta_\rho \geq n\frac{\gamma(1-\delta)}{(1-\gamma)}$$

which gives an iteration complexity under the result of [38] for an $\varepsilon$-optimal policy that is

$$n\frac{\gamma(1-\delta)}{(1-\gamma)}\log\frac{2}{(1-\gamma)\varepsilon}.$$

Recall that $n = |\mathcal{S}|$, so this iteration complexity scales linearly with the size of the state space.

## H.2   Family of MDPs on which sub-optimality gaps can be made arbitrarily small

We present how to construct a family of MDPs on which $\Delta^k(s)$ defined in Section 4 can be made arbitrarily small.

Consider an arbitrary MDP $\mathcal{M}$ with state space $\mathcal{S}$ and action space $\mathcal{A}$. For each state-action pair $(s,a) \in \mathcal{S} \times \mathcal{A}$, create a duplicate action $a'$ s.t the transitions from that action in that state are the same as for the original pair, i.e

$$p(s'|s,a) = p(s'|s,a') \quad \forall s' \in \mathcal{S}$$

and the reward is shifted down by $\delta > 0$ from the original reward, i.e

$$r(s,a') = r(s,a) - \delta.$$

This results in a new MDP $\mathcal{M}'$ with an augmented action space $\mathcal{A}'$, that is twice the size of the action space of the original MDP $\mathcal{M}$. In terms of action-value of an arbitrary policy $\pi$, this results in

$$Q_{\mathcal{M}'}^\pi(s,a) - Q_{\mathcal{M}'}^\pi(s,a') = \delta,$$

where the notation $Q_{\mathcal{M}'}^\pi$ refers to action-values in the MDP $\mathcal{M}'$. In terms of sub-optimality gaps, this gives

$$\Delta^\pi(s) \leq \delta.$$

Choosing $\delta$ small enough, we can make the step-size of [22] arbitrarily large, at least in early iterations. The step-size condition (5) of Theorem 4.1 will be less affected by this issue as it does not depend directly on $\Delta^k(s)$, and not at all in the first iteration. Beyond its generality to PMD, this illustrates the benefit of our result restricted to NPG over the result of [22].

