# OpenReview forum: "Optimal Convergence Rate for Exact Policy Mirror Descent in Discounted Markov Decision Processes"
_NeurIPS.cc/2023/Conference — NeurIPS 2023 poster_

### Official Review · Reviewer_GGgc · 2023-07-06

**Soundness:** 4 excellent
**Presentation:** 4 excellent
**Contribution:** 3 good
**Rating:** 6
**Confidence:** 4

**Summary:**

This paper proposes a new analysis of the policy mirror descent algorithm and achieves a linear convergence rate with exact $Q$-function that does not contain any distribution mismatch coefficient. Furthermore, it also proposes lower bounds showing that the linear convergence rate is optimal under specific setting and policy-dependent learning rates are necessary to achieve this rate.

**Strengths:**

- The convergence result for policy mirror descent is strong while the analysis is concise.
- The proposed two lower bounds are also considered to be novel.

**Weaknesses:**

One weakness of this paper lies in the lower bound stated in Theorem 4.2 only holds for $k<n$, which does not rule out possibility of stronger convergence rate when $k\geq n$. Therefore, it seems not very appropriate to claim that this result is optimal without any qualifiers.

Another weakness is on the lack of experiments. Since the major results of this paper strongly rely on a delibrately designed learning rate, it will be much better to empirically (in some tabular MDPs) see that this choice of learning rate is indeed better than other choices.

**Questions:**

- What are the difficulties of extending the result in Theorem 4.2 to the setting of $k\geq n$?
- Is that possible to strengthen the Theorem 4.3 in a form saying that (under the same or similar setting for $n$, $\gamma$ and $\mathcal{S}$) policy-dependent learning rates are necessary whenever we hope to achieve $\Vert V^\star-V^k\Vert_{\infty}\leq c_0\gamma^k(\Vert V^\star-V^0\Vert_{\infty}+c_1)$ for any $c_0, c_1>0$?
- Is $\eta_k$ always guaranteed to be finite?

**Limitations:**

The limitations are not addressed very well as explained in the weakness section.

---

> ### Author Rebuttal · Authors · 2023-08-09
>
> We thank reviewer GGgc for their valuable comments and feedback. We have now performed numerical experiments to validate our theoretical findings (see “global” Author Rebuttal and attached pdf), which we will include in the final version.
>
> &nbsp;
>
> ### D.1 What are the difficulties of extending the result in Theorem 4.2 to the setting of $k<n$ ?
>
> See Section C.3 (in reply to TTxd) for a discussion of this.
>
> &nbsp;
>
> ### D.2 It seems not very appropriate to claim that this result is optimal without any qualifiers.
>
> See Section C.4 (in reply to TTxd) for a discussion of this.
>
> &nbsp;
>
> ### D.3 Another weakness is on the lack of experiments.
>
> **We have now performed numerical experiments** to validate our theoretical findings (see “global” Author Rebuttal and attached pdf), which we will include in the final version.
>
> &nbsp;
>
> ### D.4 Is it possible to strengthen Theorem 4.3 ?
>
> It is possible relax the condition in Theorem 4.3 slightly. If we are hoping to achieve a bound of the form
> \begin{align*}
>     \|\|V^\star - V^k\|\|\_\infty \leq c_0 \gamma^{k} (\|V^\star - V^0\|_\infty + c_1)
> \end{align*}
>
> for $c_0, c_1 > 0$. Then to obtain a similar statement to Theorem 4.3, we will require at least:
> \begin{align*}
>     c_0 (\gamma + c_1) < 1.
> \end{align*}
>
> This condition arises from technical conditions in the proof. There is limited scope for improving the value of $c_0$, which is constrained to be bounded by $1/\gamma$ as $c_1$ approaches zero. However, if one chooses to decrease $c_0$, then larger values of $c_1$ can be accommodated. We are happy to provide more details on the steps of the proof that require this condition.
>
> &nbsp;
>
> ### D.5 Is $\eta_k$ always guaranteed to be finite?
>
> **The step-size is always finite**. It can be unbounded when $h$ is of Legendre type on the relative interior of $\Delta(\mathcal{A})$ (as for the negative entropy). Then all iterates will belong to the relative interior of $\Delta(\mathcal{A})$ and the Bregman divergences are well defined so the step-size is finite. If $h$ is of Legendre type for a domain that is larger than $\Delta(\mathcal{A})$ (as for the squared euclidean norm) then iterates will not always be in the relative interior of $\Delta(\mathcal{A})$ but there are no issues evaluating the bregman divergence for policies on the boundary of $\Delta(\mathcal{A})$ so the step-size is also finite.

---

> > ### Comment · Reviewer_GGgc · 2023-08-11
> > **Response**
> >
> > Thank you very much for your rebuttal and most of my concerns have been addressed.
> >
> > However, the optimality in Theorem 4.2 still feels questionable to me. In particular, if I didn't make any mistake, PI can converge within finite number of iterations mainly because we only have finite number of deterministic policies in a tabular MDP. Nevertheless, the number of policies scales in an order of $A^S$, which is much larger than the number of iterations guaranteed in Theorem 4.2, where $n$ is close to the number of states. That is, even though PI can achieve the optimal policy within finite number of iterations, this number of iterations may be much larger than $n$. Meanwhile, just as shown in your additional experiments, potential super-linear convergence is possible when the number of iterations is large.
> >
> > Therefore, although I acknowledge that Theorem 4.2 is a valuable result, whether it really captures the optimality of PMD still remains questionable for me.

---

> > > ### Author Response · Authors · 2023-08-11
> > >
> > > We thank reviewer GGgc for their response.
> > >
> > > While the number of policies is in the order of $|\mathcal{A}|^{|\mathcal{S}|}$, the ​**finite iteration convergence of PI only scales linearly with $|\mathcal{S}|$** [33, Theorem 3], as does our Theorem 4.2. Therefore the ​**number of iterations guaranteed by Theorem 4.2 has the same dependence on $|\mathcal{S}|$ as the number of iterations needed for exact convergence of PI**.
> > >
> > > In the final version of the paper, we will better highlight the **non-asymptotic** notion of optimality we are using.

---

> > > > ### Comment · Reviewer_GGgc · 2023-08-11
> > > > **Response**
> > > >
> > > > Thank you very much for the further clarification and now my concern has benn resolved!

---

### Official Review · Reviewer_TTxd · 2023-07-06

**Soundness:** 3 good
**Presentation:** 3 good
**Contribution:** 2 fair
**Rating:** 6
**Confidence:** 2

**Summary:**

This paper studies the convergence rate of a general family of policy gradient methods called Policy Mirror Descent (PMD) in the setting of discounted MDP, unregularized objective, and tabular policy model. In prior work ((Xiao, 2022), [7] in the paper), it has been known that with a geometrically increasing step size schedule, all PMD algorithms enjoy nice convergence rate and sample complexity with respect to the discounting factor $\gamma$, but the bounds proved in [7] also depend on a "distribution mismatch coefficient" term. This paper shows that with a more adaptive step-size schedule, this "distribution mismatch coefficient" term can be removed from the bounds. The results are derived by simplifying the analysis in [7] with a value improvement property of PMD algorithms (Lemma A.2). The paper also presents theorems to argue that the convergence rate they proved is tight, and that the adaptive step-size schedule is necessary for achieving this rate.

**Strengths:**

1. This paper seems to be a solid follow-up work after (Xiao, 2022) [7]. The paper gives a nice analysis to the PMD algorithms (removing unnecessary and "ugly" terms in previous results, simplifying the analysis framework, providing optimality/necessity arguments, etc.).

2. The results presented in this paper are nicely compared and contrasted with previous works.

**Weaknesses:**

1. I am not very sure about the significance of the results presented by the paper. While it is nice to see that with carefully scheduled step sizes every PMD algorithm could match the performance of PI and NPG, the step-size schedule required does not look very practically feasible to me, and it's still not clear if any concrete and practical PMD algorithm can demonstrate better performance than PI and NPG. Because PI and NPG are two special cases of PMD, bounds applying to all PMD algorithms are necessarily subject to the bounds of PI and NPG, so I wonder if aiming at proving bounds that "universally" apply to all PMD algorithms, as this paper does, can be truly helpful in identifying *better* PMD algorithms. Maybe the authors can consider to better motivate this work.

2. I have some concern about whether Theorem 4.2 can indeed establish the optimality of the convergence rate as proved in the paper. If I understand correctly, Theorem 4.2 says that for any given state-space size $|S|$, there exists a MDP with $|S|$ states such that no PMD algorithm can be faster than the linear $\gamma$-rate in the first $|S|/2$ PMD iterations. But this result does not bound the asymptotic performance of PMD, right? As the paper already mentioned (Line 265), PI can converge exactly to the optimal policy in finite iterations, this fact means PI, as one instance in the PMD family, must converge faster than the linear $\gamma$-rate after the first $|S|/2$ iterations. In this case I am not sure why the linear bound in Theorem 4.1 must be optimal.

**Questions:**

A. Table 1: Why isn't "linear $\gamma$-rate" checked for [7], while at Line 31 you said "*linear convergence of PMD was established by [7]*"?

B. Line 264-266: I wonder if extending (7) to k>n will necessarily contradict with the fact that PI converges in finite iterations. Specifically, the latter implies that for large enough k, we have $||V^*-V^k||=0$. To have (7) hold for such k, we "only" need to have $\gamma^k ||V^*-V^0||\leq \frac{2\delta}{1-\gamma}$, which does not look impossible to me unless $\delta$ and $V^0$ have some special constraints -- can you explain a bit more here? Although (7) is linear to $\gamma^k$, does the negative constant term $-\frac{2\delta}{1-\gamma}$ in (7) actually allows exact convergence to $V^*$?

C. Line 308: $(s^0_t, a^0_t)$ should be $(s^i_0, a^i_0)$ ?

**Limitations:**

Yes.

---

> ### Author Rebuttal · Authors · 2023-08-09
>
> We thank reviewer TTxd for their valuable comments and feedback. We have now performed numerical experiments to validate our theoretical findings (see “global” Author Rebuttal and attached pdf), which we will include in the final version.
>
> &nbsp;
>
> ### C.1 I am not very sure about the significance of the results presented by the paper.
>
> The purpose of this work is not to demonstrate that certain instances of PMD outperform PI or NPG in the exact setting. In fact, as indicated by the lower-bound in Theorem 4.2, **no instance of PMD can surpass the performance of PI in this setting**. Instead, our focus lies in understanding the theoretical properties of PMD, as discussed in more detail in section A.1 of this rebuttal (in response to wj3u). Specifically, **our results characterize the step-size required for each mirror-map to match the performance of PI**. In doing so, we develop a **novel theoretical proof scheme** that avoids visitation distributions and the use of the performance difference lemma, an important theoretical tool central to the analysis of PMD in previous work [7, 9, 17]. The **technique we use is not restricted to PMD and may be of independent interest to RL algorithms in general**.
>
> **Our results rule out searching for instances of PMD that may improve on PI or NPG**. Although proving bounds that universally apply to all PMD instances will not directly help in identifying better PMD algorithms, our work show that, in the exact setting, the algorithmic differences between instances of PMD do not affect the performance of the algorithm beyond the step-size condition.
>
> Regarding the feasibility of the step-size, a detailed discussion can be found in section B.4 (in response to YE79).
>
> &nbsp;
>
> ### C.2 Question A
>
> In Table 1, linear $\gamma$-rate is checked if the result is not only linear convergence but linear convergence with the $\gamma$-rate. [7] establish linear convergence of PMD but not with the $\gamma$-rate (see lines 217-231 for more details).
>
> &nbsp;
>
> ### C.3 Question B
>
>
> In the statement of Theorem 4.2, $\delta$ can take any value in $(0, (1-\gamma)\gamma^n)$. We could choose $\delta = \frac{1}{2} (1-\gamma) \gamma^n \|\|V^{\star} - V^0\|\|\_\infty$ (in the construction of the MDP we have $\|\|V^{\star} - V^0\|\|\_\infty < 1$ so this is a valid choice of $\delta$) and get the following bound
> \begin{align}
>     \|\|V^{\star} - V^k\|\|\_\infty \geq \Big(\gamma^{k} - \gamma^n \Big) \|\|V^{\star} - V^0\|\|\_\infty,
> \end{align}
>
> which would hold for all $k$. However, the bound would be **meaningless** for $k \geq n$ and the implication of the result would not be different to restricting $k \leq n$.
>
> In general, we can take $\delta$ arbitrarily close to 0 and the lower-bound is arbitrarily close to $\gamma^{k} \|\|V^{\star} - V^0\|\|\_\infty$.
>
> For further comments on the extension of Theorem 4.2 to arbitrary iterations, refer to the section below (C.4).
>
> &nbsp;
>
> ### C.4 I have some concern about whether Theorem 4.2 can indeed establish the optimality of the convergence rate as proved in the paper.
>
> The understanding of TTxd of Theorem 4.2 is correct, it does not bound the asymptotic performance of PMD. But any attempt to establish a lower-bound on the **asymptotic** rate of convergence for PMD (with **arbitrary** step-size) will fail.
>
> In fact, any attempt to establish a lower-bound on the convergence rate of PMD will necessarily fall into one of two categories: (i) being limited to finite iterations, as we have done in Theorem 4.2, or (ii) having the bound become negative after a finite number of iterations, as TTxd has suggested. This stems from the **exact** convergence of PI within a finite number of iterations [33].
>
> However, this has to be the case **even if we exclude PI** from the class of considered algorithms because we state Theorem 4.2 for PMD with **arbitrary** step-size. As the step-size tends to infinity, any PMD update recovers a PI update. This implies that general PMD can be arbitrarily close to PI's exact convergence for the same finite number of iterations. Thus, any lower-bound on the convergence of PMD must meet one of the above requirements ((i) or (ii)).
>
> Also, note that the finite-time convergence of PI depends on the size of the state space and action space. This **justifies having a finite-iteration lower-bound that depends on the size of the state space as ours does**. One possible improvement to Theorem 4.2 could involve having the iterations for which it holds align precisely with PI's exact finite-iteration convergence. Beyond this, Theorem 4.2 is unimprovable (given that we consider PMD for any step-size).
>
> **Rate-Optimality**
>
> Theorem 4.2 establishes that the **$\gamma$-rate is optimal in the first $|\mathcal{S}|/2$ iterations**. The explanations in the discussion above (C.4) establish that the **optimal rate is only optimal for a finite number of iterations**, and specifically up until the exact convergence of PI.
>
> With this in mind, (we believe that) the **claim of rate-optimality is valid** because for any finite number of iterations $K$, there is an MDP of size $2K + 1$ such that no PMD algorithm is faster than the linear $\gamma$-rate in those $K$-iterations. Additionally, if we consider PMD on a tabular MDP with fixed state space size $|\mathcal{S}|$, then the $\gamma$-rate is optimal for the initial $|\mathcal{S}|/2$ iterations. We will however clarify these points more explicitly in the final version.
>
> &nbsp;
>
> ### C.5 Question C
>
> Reviewer TTxd is correct. Thanks for pointing this out.

---

> > ### Comment · Reviewer_TTxd · 2023-08-17
> >
> > Thanks for the rebuttal which answered my questions in the review.
> >
> > Regarding my main concern 2, I think your argument makes sense. I now acknowledge that your Theorem 4.2 does establish an optimal convergence rate for any given iteration budget, so I consider concern 2 (as a concern on technical soundness) largely resolved. However, in light of the subtlety here, I wonder if the $\gamma$-rate is the proper way to characterize the convergence speed of PMD algorithms, given that exact convergence in finite time is expected.
> >
> > Regarding my main concern 1, now I see that the study of PMD algorithms (beyond NPG) is motivated by their *potential* advantage when function approximator is used. It is then important that the analysis in the exact setting (which may be of limited importance for its own sake) can inspire the search for better PMD algorithms in the inexact setting. Section 5 seems to be a relevant demonstration here, although the resulting performance bound is not really better yet. Overall I can better appreciate the significance of this work now.

---

> > > ### Author Response · Authors · 2023-08-18
> > >
> > > We again thank author TTxd for their response.
> > >
> > > **Characterizing the convergence speed of PMD algorithms:** . The exact convergence is in a finite number of iterations that depends on $\mathcal{S}$ and $\mathcal{A}$ and so cannot be characterised by dimension-independent bounds. Given that we seek dimension-independent bounds and want to understand convergence in the non-asymptotic regime, we need more refined bounds. Our Theorem 4.1 shows a non-asymptotic convergence bound that is dimension-independent (does not depend on $|\mathcal{S}|$ or $|\mathcal{A}|$), and together with Theorem 4.2, shows that we cannot improve on the $\gamma$-rate. This implies there is no better way of characterizing the dimension-independent convergence speed for this type of bound.
> > >
> > > Beyond the analysis in the exact setting inspiring the search for better PMD algorithms in the inexact setting, understanding the performance of PMD in the exact setting is interesting in its own right because it is a fundamental setting and as laid out in C.1, it rules out searching for instances of PMD that may improve on PI or NPG.

---

> > > > ### Comment · Reviewer_TTxd · 2023-08-22
> > > > **Post-rebuttal update**
> > > >
> > > > I raised my evaluation score to 6 because both of my two main concerns are partially resolved through the discussion. However, I retain my low confidence score (=2) as I still have conservative opinions on both concerns (see my last comment).

---

### Official Review · Reviewer_YE79 · 2023-07-06

**Soundness:** 3 good
**Presentation:** 4 excellent
**Contribution:** 3 good
**Rating:** 6
**Confidence:** 4

**Summary:**

This work considers the class of Policy Mirror Descent (PMD) methods and establishes a linear $\gamma$-rate which is dimension-free using adaptive step sizes. In particular, the convergence rate does not depend on visitation distribution mismatch coefficients or the state/action space sizes. It is further shown that this rate is optimal for PMD methods as it is the case for Policy Iteration via a matching lower bound. Adaptive step sizes are also shown to be necessary for this result to hold. Improving over prior work, the analysis which does not make use of the performance difference lemma is further extended to the inexact setting where the action value function needs to be estimated under a generative model.

**Strengths:**

- The independence of the rate on the visitation distribution mismatch coefficients is interesting and the novel analysis avoids using the performance difference lemma to achieve this. The flexibility of the analysis offers the possibility to close the gap between the convergence rates of PG methods and value-based methods in terms of $1/(1-\gamma)$ and match the known lower bound.

- The convergence rate which holds for the $l_{\infty}$ norm is shown to be optimal and the adaptive step size is also necessary.

- The paper is well-written and the contributions are clearly presented.

- The results are sound and the proof of the main result is clean.

**Weaknesses:**

- The interesting dimension-free rate seems to come with the price of more complicated adaptive step sizes compared to the increasing steps in [7].

- Some recent related works going beyond the tabular setting are missing in the discussion, especially regarding l. 131-133.

Yuan, R., Du, S. S., Gower, R. M., Lazaric, A., & Xiao, L. Linear convergence of natural policy gradient methods with log-linear policies. ICLR 2023

Alfano, C., & Rebeschini, P. (2022). Linear convergence for natural policy gradient with log-linear policy parametrization. arXiv preprint arXiv:2209.15382.

Alfano, C., Yuan, R., & Rebeschini, P. (2023). A novel framework for policy mirror descent with general parametrization and linear convergence. arXiv preprint arXiv:2301.13139.

- While a matching lower bound is provided for the convergence rate, the dependence with respect to $1/(1-\gamma)$ does not improve over prior work and does not match the known lower bound as acknowledged by the paper.

-  Leveraging a connection with Policy Iteration, a dimension free rate was previously established in the literature in [9] for the NPG method with adaptive step sizes which is a particular case of the PMD class of methods. The present work extends somehow this result with adaptive step sizes which are independent on the minimal sub-optimality gap, using a different analysis (as discussed in l. 237 to 253). The extension to the inexact setting follows the same lines as prior work [7].

**Minor:**

- l. 174: The function $h$ is continuously differentiable on the relative interior of the simplex $\Delta(\mathcal{A})$, so I guess $\pi'$ belongs to the relative interior of $\Delta(\mathcal{A})$.

**Typos:**
- l. 184: mirror map h -> $h$
- Eq. (4) (and Eq. (9)): $p \in \Delta(\mathcal{A})$ and then $p_s$ is used in the argmin.


**Questions:**

1.  How easy is it to compute the adaptive step sizes in (5)? Is there any computational tradeoff? The step sizes seem to potentially involve dependence on the state/action space sizes and would require some computational efforts to be computed. For instance, computing the set $\mathcal{A}_s^k$ requires screening the action space. Could you comment on this and compare to the step sizes in [7] regarding this aspect?

2. Does Theorem 4.3 about adaptive step-size necessity only hold for the particular case of the negative entropy for $h$ unlike all other results which hold for the general class of PMD methods? Could you comment on the reasons for that?

3. In Eq. (8), is the definition of $\tilde{\pi}_{s_i}^{k_i + 1}$ the same as in (5)? What are the states ${s_i}$ in l. 289?


**Limitations:**

The suboptimality in terms of the dependence on $1/(1-\gamma)$ (which does not match the known lower bound) is discussed both in l. 330-332 and in the conclusion.

---

> ### Author Rebuttal · Authors · 2023-08-09
>
> We thank reviewer YE79 for their valuable comments and feedback and for pointing our typos. We have now performed numerical experiments to validate our theoretical findings (see “global” Author Rebuttal and attached pdf), which we will include in the final version.
>
> &nbsp;
>
> ### B.1 Weaknesses
>
> - 1. As shown in Theorem 4.3, this step-size is in fact **necessary** for the dimension free $\gamma$-rate. In section B.2 below, we also discuss the computational feasibility of the step-size.
> - 2. Thanks for mentioning these works. We will include them in the discussion.
> - 3. We stress that the focus of our work is on the **exact setting** (Theorem 4.1, 4.2 and 4.3). We include the inexact setting to showcase that **our analysis can also be used in this setting to remove an instance dependent factor** (that may depend on dimension) that appeared in prior work [7]. More generally, the analysis can easily be combined with any scheme for estimating the Q functions (see Section 5), paving the way for further improvements in instance-independent sample complexity results should more efficient estimation procedures be developed.
> - 4. While a dimension free rate was established in [9] for the specific case of NPG with adaptive step size, **no justification was provided for this step-size until our Theorem 4.3**.
>
> **Minor:** If $h$ is of Legendre type on the relative interior of $\Delta(\mathcal{A})$ (as for the negative entropy), then all iterates will belong to the relative interior of $\Delta(\mathcal{A})$ as long as $\pi^0$ does. But if $h$ is of Legendre type for a domain that is larger than $\Delta(\mathcal{A})$ (as for the squared euclidean norm) then iterates will not always be in the relative interior of $\Delta(\mathcal{A})$, nor do we require $\pi^0$ to be in the relative interior of $\Delta(\mathcal{A})$.
>
> &nbsp;
>
> ### B.2 Question 1
>
> Here are some additional comments regarding the adaptive step-size condition. We will include these in the final version. Recall $\mathcal{A}^k_s = \\{a\in\mathcal{A}:Q^k(s,a)=\max_{a'\in \mathcal{A}}Q^k(s,a')\\} $ denotes the set of optimal actions in state $s$ under policy $\pi^k$ and $\widetilde{\Pi}^{k+1}\_s$ is the set of greedy policies w.r.t $Q^k_s$ in state s, i.e $\widetilde{\Pi}^{k+1}\_s = \Big \\{ p \in \Delta(\mathcal{A}) :\sum_{a\in\mathcal{A}^k\_s}p(a)=1\Big\\}$. For $\\{c_k\\}\_{k \in \mathbb{Z}\_{\geq 0}}$ a sequence of positive reals, the step-size condition in (5) is
> \begin{equation*}
>         \eta_k\geq\frac{1}{c_k}\max\_{s\in \mathcal{S}}\Big\\{\min_{\widetilde{\pi}^{k+1}\_s\in \widetilde{\Pi}^{k+1}\_s} D_h(\widetilde{\pi}^{k+1}\_s,\pi^k\_s)\Big\\}.
> \end{equation*}
>
> **The dependence on the action space can be removed:** The condition in (5) is a minimum over $\widetilde{\pi}^{k+1}_s\in \widetilde{\Pi}^{k+1}_s$ so taking the condition for any $\widetilde{\pi}^{k+1}_s \in \widetilde{\Pi}^{k+1}_s$ will be sufficient. In particular, we used the minimum to have the smallest condition but that is not necessary. Also note that in many cases, there will only be one greedy action, i.e. $\mathcal{A}^k_s$ will consist of a single action and there will just be one policy in $\widetilde{\Pi}^{k+1}_s$.
>
> As for the maximum over $s \in \mathcal{S}$, this is imposed because we are using the same step-size in all states for simplicity. If we allow different step-size in each state, then the step-size in state $s$, denoted $\eta_k(s)$ would just have to satisfy
> \begin{align*}
>         \eta_k(s)\geq\frac{1}{c_k}\Big\\{\min_{\widetilde{\pi}^{k+1}_s\in \widetilde{\Pi}^{k+1}_s}D_h(\widetilde{\pi}^{k+1}_s,\pi^k_s)\Big\\}.
> \end{align*}
>
> Furthermore if we choose any $\widetilde{\pi}^{k+1}_s \in \widetilde{\Pi}^{k+1}_s$, then the following is sufficient:
> \begin{align*}
>         \eta_k(s)\geq\frac{1}{c_k}D_h(\widetilde{\pi}^{k+1}_s,\pi^k_s).
> \end{align*}
>
> **The computational complexity of computing the step-size is then that of computing the Bregman divergence between two policies.**
>
> So in general, this will be more expensive than the step-size of [7]: $\eta_k = \eta_0 / \gamma^k$. However, in many cases, computing the Bregman divergence is no more expensive than computing the PMD update in a state given the step-size (e.g. for relative-entropy and squared euclidean distance mirror maps). If so, the cost of computing the step-size in (5) adds to the overall computational complexity of the algorithm but the order of magnitude would remain the same.
>
> &nbsp;
>
> ### B.3 Question 2
>
> Yes, Theorem 4.3 only holds for the particular case of the negative entropy (NPG). This limitation is due to technical reasons rather than fundamental ones. The **proof relies on the closed form update** of NPG and we don't have this for general mirror map.
>
> We need the closed form update because we show that there is an iteration $k_0$ where the amount of probability mass on the sub-optimal action of a state has to drop by a certain amount from $\pi^{k_0}$ to $\pi^{k_0 + 1}$ in order to satisfy the $\gamma$-rate condition of Theorem 4.3. We then explicitly relate this drop to the size of the step-size by using the closed form update.
>
> &nbsp;
>
> ### B.4 Question 3
>
> In Eq. (8), the definition of $\tilde{\pi}^{k_i+1}\_{s_i}$ is the **same** as in (5). In particular, the MDP considered in the proof has 2 actions in each state so it is the policy which is greedy w.r.t $Q^{\pi^{k_i}}_{s_i}$.
>
> As for the states in l. 289, the MDP consists of a chain of $n+1$ states, $n$ of which have an additional state associated to them (total of $2n + 1$ states). The states $\{s_1,...,s_n\}$ in Theorem 4.3 are the $n$ states in the chain which have this additional state (see Appendix E for more detail on the construction of this MDP). In fact, these states are the only none-trivial states of the MDP where the 2 actions are not both optimal. **The main point is that this necessary condition is enforced in $n$ distinct iterations in the $n$ distinct non-trivial states of the MDP.**

---

> > ### Comment · Reviewer_YE79 · 2023-08-14
> > **post rebuttal**
> >
> > I thank the authors for their detailed responses to my questions and relevant clarifications as well as their additional experiments which illustrate in practice the behavior of the adaptive steps on the hard provided instance. I also went through the other reviews and responses and I maintain my original positive score.

---

### Official Review · Reviewer_wj3u · 2023-07-07

**Soundness:** 2 fair
**Presentation:** 2 fair
**Contribution:** 2 fair
**Rating:** 5
**Confidence:** 1

**Summary:**

The paper studies the convergence rate of Policy Mirror Descent. The authors prove a $\gamma$-rate for exact PMD update and a matching lower bound for this class of algorithm. Then they prove a sample complexity bound of $O(\epsilon^{-2}(1-\gamma)^{-8})$.



**Strengths:**

The paper analyses an interesting problem on the convergence rate for a general class of RL algorithms. The work is nicely structured and the theoretical results seems not trivial to obtain.

**Weaknesses:**

In the reviewer's opinion the paper does not convey clearly the contribution of the paper. More specifically it's hard to understand if the paper advances theoretically the field of RL or just the theoretical understanding of PMD.

Minor:
The following sentence in the abstract: "Motivated by the instability of policy iteration (PI) with inexact policy evaluation, unregularised PMD algorithmically regularises the policy improvement step of PI without regular5 ising the objective function" sounds convoluted and maybe even grammatically incorrect. Please have a look

**Questions:**

1) A lower bound on a class of algorithm and not on the problem seems strange. Could please the author elaborate on this point? Is this standard in RL literature?
2) The terminology: Unregularized PMD is not explicitly explained in text. It would be helpful to clarity it explicitly.

**Limitations:**

yes

---

> ### Author Rebuttal · Authors · 2023-08-09
>
> We thank reviewer wj3u for their valuable comments and feedback. We have now performed numerical experiments to validate our theoretical findings (see “global” Author Rebuttal and attached pdf), which we will include in the final version.
>
> &nbsp;
>
> ### A.1 In the reviewer's opinion the paper does not convey clearly the contribution of the paper. More specifically it's hard to understand if the paper advances theoretically the field of RL or just the theoretical understanding of PMD.
>
> The contributions of our work can be summarised as
> - **1. Optimal Rate:** We show that exact PMD achieves the linear $\gamma$-rate (Theorem 4.1) and that this is optimal for any mirror map (we give a lower bound in Theorem 4.2).
> - **2. Step-Size Necessity:** We show that adaptivity in the step-size is necessary for the optimal $\gamma$-rate (Theorem 4.3; in at least one instance of PMD). The instance is PMD with negative-entropy and prevents improvement on the step-size in Theorem 4.1 for general mirror map.
> - **3. Simple Analysis:** Our analysis (Section 6) avoids visitation distributions and the performance difference lemma, which is one of the most widely used proof techniques to analyse PMD-like algorithms (c.f. below).
> - **4. Sample Complexity:** We extend our results to inexact PMD (Theorem 5.1) giving the first result optimal with respect to dimension ($|\mathcal{S}|, |\mathcal{A}|$) for PMD under a generative model. In particular, we remove an instance dependent factor (that may depend on dimension) that appeared in prior work [7]. More generally, the analysis can easily be combined with any scheme for estimating the Q functions (see Section 5), paving the way for further improvements in instance-independent sample complexity results should more efficient estimation procedures be developed.
>
> These contributions advance the theoretical understanding of PMD. Algorithms achieving the $\gamma$-rate were already known (Policy Iteration / Value Iteration) but **it was not known for PMD** in general prior to our work. In particular, we characterise both **necessary** (Theorem 4.3) and **sufficient** (Theorem 4.1) conditions for PMD to achieve the $\gamma$-rate. We also **improve the sample-complexity** of PMD under a generative model. These are advances in the theoretical understanding of PMD specifically.
>
> The **novelty in our analysis advances the field of RL** as a whole. Our analysis avoids the use of the  performance difference lemma, an important theoretical tool central to the analysis of PMD in previous work [7, 9, 17]. The technique we use is not restricted to PMD and is **applicable to RL algorithms in general**.
>
> See C.1 (in the response to TTxd)  for more discussion on the significance of the contributions.
>
> &nbsp;
>
> ### A.2 Sentence in abstract sounds convoluted and maybe even grammatically incorrect.
>
>
> This sentence was originally trying to convey both the distinction between Policy Iteration (PI) and PMD as well as the distinction between PMD on a regularised and unregularised MDP. The important distinction is between PI and PMD. As a result, we have simplified the sentence to the following:
>
> **“Motivated by the instability of policy iteration (PI) with inexact policy evaluation, PMD algorithmically regularises the policy improvement step of PI".**
>
> As for the second point regarding PMD on a regularised and unregularised MDP, we will now make an explicit remark about this in the main text (see A.4).
>
>
>
>
> &nbsp;
>
>
> ### A.3 A lower bound on a class of algorithm and not on the problem seems strange.
>
> We emphasise that the main contribution of the work is on the $\gamma$-rate upper bound for PMD (Theorem 4.1). However, to demonstrate the tightness/optimality of this rate, we also provide a lower bound (Theorem 4.2).
>
> This type of lower-bound is not standard in the RL literature. However, lower bounds for solving infinite-horizon MDPs with knowledge of the underlying dynamics are rare, and the only work we are aware of, by Chen and Wang (https://arxiv.org/abs/1705.07312), focuses on the computational complexity, which is not our focus.
>
> We assume access to a policy-evaluation oracle for any policy. The problem we consider is the number of iterations needed to find an $\varepsilon$-optimal policy, where at each iteration we are allowed one query from our oracle for any policy. As far as we know, there is currently no existing lower bound for this problem.
>
> It is unclear what the lower-bound on the problem in general might be. However **our lower-bound holds on this problem for a large class of algorithms**: any algorithm that at each iteration increases the probability of the greedy action with respect to the action-value of the previous policy.
>
> In addition, as we remarked in A.1, we are interested in the theoretical understanding of PMD, so we **want to compare the rate we establish for PMD to the best rate possible for PMD**. Our algorithmic specific lower-bound answers this question **positively**.
>
>
>
>
>
>
> &nbsp;
>
>
> ### A.4 Unregularized PMD is not explicitly explained in text.
>
> In an attempt to avoid any confusion, we used the terminology “unregularized PMD” in the abstract to refer to the version of PMD we consider (as opposed to a different version used in prior work). For clarity, we will not use this terminology (see A.2) and will make the following explicit remark when defining PMD in Section 3.1:
>
> **“The update (4) of PMD considered in this work uses the true action-value $Q^\pi$. In prior work, PMD has sometimes been applied to regularised MDPs [11] where the action-value is augmented with some form of regularisation and is no longer the true action-value. This is a different algorithm that converges to a policy that is not optimal in the original unregularised MDP.”**

---

### Author Rebuttal · Authors · 2023-08-09

We have now performed numerical experiment to validate our theoretical findings (see attached pdf), which we will include in the final version.

The plots in the attached PDF are made by running NPG (PMD with negative-entropy mirror map, see [9]) on the hard MDP used to prove Theorem 4.2. We use $n = 25$ so $|\mathcal{S}| = 51$ and $|\mathcal{A}| = 2$. We use $\delta = (1-\gamma) \gamma^n / 100$, $\gamma = 0.99$, $\pi^0(a_1 | s) = \alpha$ (note that $a_1$ is the optimal action in each state). In each of the plots, we compare the performance of NPG using our step-size (5) from Theorem 4.1 (denoted **Adaptive** in the plots and by ADA in the discussion below) with $c_k = \gamma^{2k}$ and $c_0 = 1$ and the step-size of [7]: $\eta_k = \eta_0 / \gamma^k$ (denoted by INC in the discussion below) for a fixed $\eta_0 = 1$. In each of the plots, the left plot is $\|\|V^\star - V^k\|\|_\infty$ against iterations and the right plot is $\eta_k$ against iterations.

In all plots, the green curve represents the $\gamma$-rate. In particular it is the curve of $y = \gamma^x$.

&nbsp;


### Discussion:

With fixed initial step-size $\eta_0 = 1$ for INC, the **convergence of NPG with INC can be made arbitrarily slow in early iterations** by choosing $\alpha$ (= $\pi^0(a_1 |s)$) closer to 0 (Figures 1-3). The **convergence of NPG with ADA is unaffected** by small values of $\alpha$ because it adapts to them.

**These observations agree with the theoretical results we established in our paper.** Our adaptive step-size (5) achieves the $\gamma$-rate (Theorem 4.1) and this is optimal for iterations $k$ up until $n=25$ (Theorem 4.2). The adaptivity in the step-size is necessary to achieve the $\gamma$-rate in these early iterations (Theorem 4.3).

In later iterations (Figure 1: iterations $\approx 100$. Figure 2: iterations $ \approx 200$), INC has faster convergence than ADA. In these later iterations, acting greedily gives the optimal policy so INC converges faster because the step-size is larger (see Figure 1-2 right plot). This suggests **combining ADA and INC by taking the maximum of both (Figure 4), yielding faster convergence** than both. This is encouraging for future-work on super-linear convergence (as discussed in [7]) beyond the optimal $\gamma$-rate valid up to a finite iteration (recall that for arbitrary step size, PMD can be arbitrarily close to policy iteration, which converges in finite iterations; see C.4 in the response to Reviewer TTxd).

---

### Decision · Program_Chairs · 2023-09-21

**Decision:**

Accept (poster)

**Comment:**

This paper obtains the optimal dimension-free rates for Policy Mirror Descent (PMD) and shows that unregularized PMD methods with an adaptive step-size can match the performance of PI.

The reviewers agree that the paper is well-written, and its contributions merit acceptance. I agree and recommend the acceptance of this paper. Please incorporate the reviewer's comments in the final version of the paper. In particular, addressing the following concerns will help strengthen the current version of the paper:
- Adding the numerical experiments included as part of the rebuttal
- Clearly including the discussion about the computational complexity of calculating the adaptive step-size.
- Include the discussion with Rev. TTxd and GGgc about the (non)-asymptotic performance of PMD.